# Engineered Pullulan-Collagen-Gold Nano Composite Improves Mesenchymal Stem Cells Neural Differentiation and Inflammatory Regulation

**DOI:** 10.3390/cells10123276

**Published:** 2021-11-23

**Authors:** Meng-Yin Yang, Bai-Shuan Liu, Hsiu-Yuan Huang, Yi-Chin Yang, Kai-Bo Chang, Pei-Yeh Kuo, You-Hao Deng, Cheng-Ming Tang, Hsien-Hsu Hsieh, Huey-Shan Hung

**Affiliations:** 1Department of Neurosurgery, Neurological Institute, Taichung Veterans General Hospital, Taichung 407204, Taiwan; yangmy04@gmail.com (M.-Y.Y.); jean1007@gmail.com (Y.-C.Y.); 2National Defense Medical Center, Graduate Institute of Medical Sciences, Taipei 11490, Taiwan; 3College of Nursing, Central Taiwan University of Science and Technology, Taichung 406053, Taiwan; 4College of Medicine, National Chung Hsing University, Taichung 40227, Taiwan; 5Department of Medical Imaging and Radiological Sciences, Central Taiwan University of Science and Technology, Taichung 406053, Taiwan; bsliu@ctust.edu.tw (B.-S.L.); kbwork2021@gmail.com (P.-Y.K.); a0978327705@gmail.com (Y.-H.D.); 6Department of Cosmeceutics and Graduate, Institute of Cosmeceutics, China Medical University, Taichung 40402, Taiwan; patty19991014@gmail.com; 7Graduate Institute of Biomedical Science, China Medical University, Taichung 40402, Taiwan; dlovan2131@yahoo.com.tw; 8College of Oral Medicine, Chung Shan Medical University, Taichung 40201, Taiwan; hhhsu@vghtc.gov.tw; 9Blood Bank, Taichung Veterans General Hospital, Taichung 407024, Taiwan; ranger@csmu.edu.tw; 10Translational Medicine Research, China Medical University Hospital, Taichung 40402, Taiwan

**Keywords:** skin wound healing, pullulan–collagen–gold nanoparticles, mesenchymal stem cells, neuronal differentiation, anti-inflammation

## Abstract

Tissue repair engineering supported by nanoparticles and stem cells has been demonstrated as being an efficient strategy for promoting the healing potential during the regeneration of damaged tissues. In the current study, we prepared various nanomaterials including pure Pul, pure Col, Pul–Col, Pul–Au, Pul–Col–Au, and Col–Au to investigate their physicochemical properties, biocompatibility, biological functions, differentiation capacities, and anti-inflammatory abilities through in vitro and in vivo assessments. The physicochemical properties were characterized by SEM, DLS assay, contact angle measurements, UV-Vis spectra, FTIR spectra, SERS, and XPS analysis. The biocompatibility results demonstrated Pul–Col–Au enhanced cell viability, promoted anti-oxidative ability for MSCs and HSFs, and inhibited monocyte and platelet activation. Pul–Col–Au also induced the lowest cell apoptosis and facilitated the MMP activities. Moreover, we evaluated the efficacy of Pul–Col–Au in the enhancement of neuronal differentiation capacities for MSCs. Our animal models elucidated better biocompatibility, as well as the promotion of endothelialization after implanting Pul–Col–Au for a period of one month. The above evidence indicates the excellent biocompatibility, enhancement of neuronal differentiation, and anti-inflammatory capacities, suggesting that the combination of pullulan, collagen, and Au nanoparticles can be potential nanocomposites for neuronal repair, as well as skin tissue regeneration in any further clinical treatments.

## 1. Introduction

Human skin, the special interface that separates the body from the external environment, has various specific and important functions. The integrity of human skin plays a vital role in sensation, temperature regulation, and water loss prevention, while also providing protection against mechanical injuries or pathogens [1,2]. However, skin integrity can be destroyed by various physical and chemical lesions such as surgery, chronic skin ulcers, and hazardous chemical substances [3,4]. Skin wounds can be caused by acute or chronic factors such as severe burn injury, diabetic ulcers, or other mechanical injuries. Dermis contains dermal nerve fibers, and where the injury occurred, the sensory system may be lost [5]. Indeed, traditional tissue engineering has its limitations, such as the shortage of suitable organs and tissue, which can lead to the death of patients while waiting for treatment [6]. Over the past decades, skin graft engineering has been the primary clinical approach for patients with skin injuries [7]. Skin graft approaches can be classified into five categories: autografting, allografting, xenografting, culturing of autologous cells, and the usage of the extracellular matrix (ECM). Skin autografting is the standard clinical application for patients upon experiencing burn excision. However, for patients with extensive burns (over 60% of the body surface area), the lack of the patients’ own donor skin becomes a shortcoming which affects patient survival [8]. Thus, human allograft skin transplantation from deceased donors can provide temporary skin coverage until autografts are available to overlay on the injury area, as allograft skin tissue is more compatible with human skin tissue [9]. Despite the above mentioned advantages, allografts also have the risk of disease transmission and immunological rejection [10]. For xenografting, the skin tissue is acquired from other species. It had been verified that the fresh skin from frogs could enhance the wound-healing process [11], while the skin from lepidosaurians has allowed keratinocyte differentiation in epidermal layers [12]. Pig skin transplantation is still technically approved, and has also been used for wound healing [13]. As a xenogeneic skin graft material, the advantage in using pig skin is that its source is not scarce, and the inner layer of the skin is abundant with collagen, which offers a high biocompatibility. However, the disadvantage is that the skin from animals may cause immunoreactivity and should therefore be treated through specific methods. Furthermore, the treated animal skin cannot obtain a blood supply from the wound, leading to an eventual deficiency in skin adherence [14]. Moreover, culturing autologous cells in vitro can provide permanent coverage to a large, injured area. However, this requires more time and is also more expensive [9]. The usage of extracellular matrices (ECMs) such as collagen and chitosan has demonstrated less antigenicity, better biodegradability, and superior biocompatibility. Certain literature has also indicated that collagen–chitosan porous scaffolds provide a suitable microenvironment that could promote dermal regeneration and angiogenesis for tissue repair [15].

Artificial materials combined with cells such as cell-based skin substitutes and epithelial/dermal replacements materials, are well applied in skin tissue engineering [16]. Furthermore, cell-based substitutes comprising living skin cells and ECM are considered to be more effective [17]. TransCyte is a neodermis regeneration matrix composed of a nylon mesh incorporating allogeneic human dermal fibroblasts [18], and it is verified to facilitate re-epithelialization capacity and the healing period for patients suffering from partial or full-thickness skin burns [19]. Dermagraft is a degradable substitute produced by allogeneic human dermal fibroblasts and a collagen scaffold, which shows tear resistance and decreases the rate of infection for chronic diabetic foot ulcer patients [20,21]. Keratinocytes are the major cells in the process of re-epithelialization, secreting various cytokines and growth factors [22]. Keratinocyte dressing treatments demonstrate no side-effects and facilitate wound repair for non-healing diabetic neuropathic foot ulcers [23]. However, the substitutes are commonly high cost, and require storage under special conditions [17]. Stem cells such as mesenchymal stem cells (MSCs) have been widely used in clinical approaches owing to the ease with which they can be isolated, lower immune responses, high self-renewal, and differentiation capacities [24]. MSCs can differentiate into neurons, endothelial cells, osteocytes, and adipocytes due to the induction of a microenvironment having growth factors and cytokines, leading to efficiency in tissue and wound healing [25]. Our previous studies demonstrated that MSCs culturing with nanogold–collagen or nanogold–fibronectin composites exhibited the enhancement of proliferation and endothelialization, indicating MSCs can be a fascinating material for tissue repair [26,27]. Furthermore, human skin fibroblasts (HSFs) have also been extensively applied to biomedical treatments. HSFs can synthesize extracellular matrices (ECMs) such as collagen, which is the major component of human dermal tissue involved in the process of skin repair [28]. HSFs are mostly presented in dermis, which also plays a critical role in angiogenesis and wound healing [29]. However, neurons are insufficient for self-regeneration, and the materials mentioned above just form layers and do not reconstruct nerve fibers [30]. Thus, neuronal regeneration is a critical issue for skin tissue repair.

Pullulan, a natural linear homopolysaccharide produced by the fungus *Aureobasidium pullulans* [31], has been widely used as a biomaterial scaffold owing to its biodegradability, nontoxicity, anti-oxidant abilities, and excellent mechanical properties [32]. A previous study has indicated that the combination of pullulan and gelatin demonstrated the enhancement of cell proliferation for wound healing, including burn injury and chronic wounds [33]. Collagen is the major structural protein in bones, skin, and blood vessels [34], and has been widely used as a vascularization scaffold in clinical treatments. The literature has verified that collagen scaffolds have the ability to support angiogenesis and neovasculature formation [35]. A vascular endothelial growth factor (VEGF)-modified collagen scaffold could facilitate penetration and the proliferation of ECs in the scaffold [36]. Furthermore, a study demonstrated that pullulan–collagen composite hydrogels were fabricated through the salt-induced phase inversion technique. The results elucidate the recruitment of stromal cells and the formation of vascularized granulation tissue, promoting the efficiency of in vivo wound healing [37]. Additionally, gold nanoparticles (Au) are biocompatible materials that can be cross-linked with biomolecules to regulate cell behavior [38]. The Au nanoparticles can be easily synthesized and controlled in their size and shape through surface plasmon resonance (SPR) [39]. The reference identified that the fabrication of Au and silk fibers transformed into a sheet form by electrospinning to obtain conduits, and could enhance the adhesion and proliferation of Schwann cells. Furthermore, the nanocomposites did not exhibit any toxicity or inflammatory responses through in vivo assessment. Moreover, the silk–gold nanocomposite-based nerve conduit could induce neuronal differentiation and the regeneration of the peripheral nerve. Moreover, our previous study also demonstrated that Au nanoparticles fabricated with polyurethane (PU) could enhance the endothelialization of endothelial progenitor cells (EPCs), contributing to tissue repair [40].

In the current research, we prepared the nanocomposite Pul–Col–Au to investigate its potential in biological performances, including cell proliferation and neuronal differentiation capacities, as well as anti-inflammatory responses through in vitro and in vivo assays. Our aim was to explore the use of a novel nanomaterial for skin tissue regeneration.

## 2. Materials and Methods

### 2.1. Material Synthesis

#### 2.1.1. Preparation of Pullulan Solution (Pul)

First, 0.125 g, 0.25 g, 0.5 g, and 1 g of white Pul powder (MW: 20,000, Sigma-Aldrich, Burlington, MA, USA) were separately and fully dissolved in 10 mL of deionized water in 15 mL centrifuge tubes (total volume was 10 mL in each tube). In the current research, the concentration of Pul solution was 0.125 g/10 mL, 0.25 g/10 mL, 0.5 g/10 mL, and 1 g/10 mL, each respectively coated on a 96-well culture plate for 30 min. Afterwards, the residual solution was removed and the MSCs and HSFs (1 × 10^4^/well) were then cautiously seeded into the 96-well plate with Pul coatings for MTT examination at 24 and 48 h. Thus, the final concentration, 0.25 g/10 mL of Pul solution, was selected for the subsequent experiments.

#### 2.1.2. Preparation of Collagen (Col)

Type I collagen solution (4.88 mg) was purchased from BD Biosciences (Canton, MA, USA). The 1030 μL of type I Col was completely mixed with 8970 μL of deionized water. The total volume was 10 mL, with the final concentration of collagen diluted to 0.5 μg. The mixing ratio of the Col solution was calculated based on the formula M1V1 = M2V2 (M: the concentration of the solution, V: the volume of the solution).

#### 2.1.3. Preparation of Pullulan–Gold Nanoparticles (Pul–Au)

The solution of gold nanoparticles (50 ppm) was purchased from GNT Biotech & Medicals Corporation, (GNTbm, Taipei, Taiwan), with the diameter of the Au nanoparticles being approximately 3–5 nm. First, 0.125 g, 0.25 g, 0.5 g, and 1 g of white Pul powder were fully mixed with 7560 μL of deionized water, then 2440 μL of Au nanoparticle solution (12.2 ppm) was further added. The total volume was 10 mL with the mixing ratio of the Pul–Au solution calculated based on the formula M1V1 = M2V2 (M: the concentration of the solution, V: the volume of the solution).

Each solution of Pul–Au (a pullulan concentration of 0.125 g/10 mL, 0.25 g/10 mL, 0.5 g/10 mL, and 1 g/10 mL, and a gold nanoparticles concentration of 12.2 ppm) was then added onto a 96-well culture plate and coated for 30 min, respectively. After removing the residual solution, the MSCs and HSFs at the cell density of 1 × 10^4^ per well were seeded into the 96-well plate with Pul–Au coatings for MTT examination (24 and 48 h). In the end, the Pul–Au solution containing 0.25 g/10 mL of Pul and 12.2 ppm of Au nanoparticles was selected for the upcoming experiments.

#### 2.1.4. Preparation of Pullulan–Collagen (Pul–Col)

For the Pul–Col solution, 0.125 g, 0.25 g, 0.5 g, and 1 g of Pul powder were mixed with 8970 μL of deionized water, with 1030 μL of Col (diluted to 0.5 μg) then added. The total volume of the Pul–Col solution was 10 mL in each tube. For MTT examination at 24 and 48 h, the Pul–Col solution (0.125 g/10 mL, 0.25 g/10 mL, 0.5 g/10 mL, and 1 g/10 mL of Pul and 0.5 μg of Col in each solution) was also added to a 96-well culture plate for 30 min of coating. The residual solutions were discarded before the MSCs and HSFs at the cell density of 1 × 10^4^ per well were cautiously seeded onto the culture plate with the Pul–Col coatings. Hereafter, the Pul–Col solution containing 0.25 g/10 mL of pullulan and 0.5 μg of collagen was chosen for the subsequent experiments.

#### 2.1.5. Preparation of Pullulan–Collagen–Gold Nanoparticles (Pul–Col–Au)

To prepare the Pul–Col–Au solution, 0.125 g, 0.25 g, 0.5 g, and 1 g of Pul powder were mixed with 8726 μL of deionized water in each of the 15 mL centrifuge tubes. Then, 1030 μL of Col (0.5 μg after dilution) was added, along with 244 μL of Au nanoparticle (12.2 ppm) solution into each 15 mL centrifuge tube. Thus, the total volume for each Pul–Col–Au solution was 10 mL. The mixing ratio of the Pul–Col–Au solution was calculated according to the formula M1V1 = M2V2 (M: the concentration of the solution, V: the volume of the solution). Each Pul–Col–Au solution was also coated for 30 min in a 96-well plate for MTT assay at 24, 48, and 96 h, as described in the above procedure. Thus, the mixture of 0.25 g/10 mL pullulan, 0.5 μg collagen, and 12.2 ppm of AuNPs was used for the experiments which followed. 

In detail, when preparing the composite material, 244 uL of 12.2 ppm Au solution was added, and finally the whole solution was diluted to 10 mL. It can be calculated that the solution contained 0.0029768 mg of AuNPs, which can be converted into 78,471 Au/mL. Then, 0.594 mL of the above solution was added to a glass coverslip for plating. Therefore, it can be calculated that each coating contained 149,096 Au nanoparticles. The area of the glass coverslip was 1.9 cm^2^, so each square centimeter contained 78,471 Au nanoparticles. Additionally, the weight percentages of Pul, Col, and Au in the Pul–Col–Au composite were 99.9986%, 0.00020%, and 0.00119%, respectively.

#### 2.1.6. Preparation of Collagen–Gold Nanoparticles (Col–Au)

The Col–Au solution was fully mixed with 8726 μL of deionized water, 1030 μL of Col (0.5 μg after dilution), and 244 μL of Au nanoparticles (12.2 ppm), creating a total volume of 10 mL. The mixing ratio was calculated using the formula M1V1 = M2V2 (M: the concentration of the solution, V: the volume of the solution).

### 2.2. Material Characterization

#### 2.2.1. Scanning Electron Microcopy (SEM)

A scanning electron microscope (JEOL JEM-5200, JEOL Ltd., Akishima, Tokyo, Japan) was employed to observe the shape of the Pul–Col–Au nanoparticles. Image Pro software (Media Cybernetics, Burlington, MA, USA) was used to analyze the size of the nanoparticles (n = 10).

#### 2.2.2. Transmission Electron Microscope (TEM)

TEM images of the nanoparticles were obtained from a transmission electron microscope (JEM 1010, JEOL Ltd., Akishima, Tokyo, Japan). The voltage was set at 80 keV to clearly observe the size and structure of the nanoparticles. Before the observation, 5 μL of nanoparticles was suspended onto a copper-coated TEM grid and dried out at room temperature.

#### 2.2.3. Dynamic Light Scattering (DLS)

Dynamic light scattering (DLS) analysis was conducted by a Malvern Zetasizer Nano ZS instrument (Malvern Panalytical Ltd., Malvern, Worcestershire, UK) and manipulated using a light source (532 nm) with a 90° fixed scatter angle. The measurements were processed after adding 1 mL of the colloidal sample in a cuvette with a 1 cm optical path. To determine the zeta potential of the nanoparticles, the solutions were moved into transparent cuvettes. Pul–Col–Au solution was investigated through a Zetasizer Nano ZS instrument coupled with a 633 nm He–Ne laser at 25 °C for the upcoming investigations. The experiments were performed in triplicate.

#### 2.2.4. Hydrophilicity Property Examination

The hydrophilicity property of each nanocomposite was also investigated. Each sample was loaded on silicon substrates, with 0.7 μL of distilled water added dropwise onto the surface of the nanocomposites. The water contact angle of each nanocomposite was further determined through a PGX model instrument at RT in quadruplicate.

#### 2.2.5. UV-Visible Spectroscopy (UV-Vis)

The absorption spectrum of each nanocomposite was detected using a Helios Zeta UV-Vis spectrophotometer (ThermoFisher, Pittsfield, MA, USA). The range of the wavelength was from 400 to 600 nm, where 520 nm is the typical absorption wavelength of Au nanoparticles. Origin Pro 8 (OriginLab Corporation, Northampton, MA, USA) software was applied to further quantify the data.

#### 2.2.6. Fourier Transform Infrared Spectroscopy (FTIR)

The FTIR spectra was measured by a Fourier transform infrared spectrometer (Shimadzu Pretige-21, Kyoto, Japan). First, 200 mg of each nanocomposite sample was fully mixed with 0.06 g of potassium bromide (KBr, Sigma-Aldrich, Burlington, MA, USA), then dried out at 100 °C. Each sample was independently scanned eight times at a scanning range of 500–4000 cm^−1^ with a 2 cm^−1^ resolution in order to obtain the FTIR spectra.

#### 2.2.7. Surface-Enhanced Raman Spectroscopy (SERS) and X-ray Photoelectron Spectroscopy (XPS)

A Raman microscopic system (LABRAM HR UVVIS−NIR Version, HORIBA Ltd., Kyoto, Japan) was used to investigate the chemical structure of the nanomaterials. Test samples were loaded on a silicon plate before a laser wavelength of 633 nm was applied with an incident power of 20 mW, under the conditions of holographic grating, 1800 grooves mm^−1^, for an interval of 60 s. The resolution of the charge-coupled device was 1024 × 3256 pixels. The elemental composition of the test materials was evaluated through X-ray photoelectron spectroscopy (XPS) (AXIS Ultra, Kratos Analytical Ltd., Stretford, Manchester, England). The radiation was emitted from a monochromatic Mg Kα X-ray source.

### 2.3. Cell Culture

#### 2.3.1. Wharton’s Jelly-Derived Mesenchymal Stem Cells

The MSCs were harvested and isolated from a human umbilical cord’s Wharton’s jelly tissue [41]. The cells were cultured in a high glucose Dulbecco’s Modified Engle’s Medium (DMEM) (Gibco, Thermo Fisher, Waltham, MA, USA), which was supplemented with 10% FBS, 1% (*v*/*v*) antibiotics (100 U/mL P/S), and 1% sodium pyruvate. After reaching confluency under the appropriate conditions (37 °C, 5% CO_2_), the cells were collected with 0.05% trypsin-EDTA for the follow-up experiments. 

To characterize the phenotypes of the MSCs, the cells were harvested and detached with 2 mM EDTA in phosphate-buffered saline (PBS), then washed by PBS with 2% bovine serum albumin (BSA) and 0.1% sodium azide (Sigma-Aldrich, Burlington, MA, USA). Further, the cells were incubated with antibodies conjugated with fluorescein isothiocyanate (FITC) or PerCP-Cy5-5-A. The indicated markers were CD34-FITC, CD45-FITC, CD44-PE, and CD90-PerCP-Cy5-5-A (BD Pharmingen, San Diego, CA, USA). FITC-conjugated IgG1 and PerCP-Cy5-5-A-conjugated IgG1 (BD Pharmingen) were demonstrated as isotype controls. The specific surface antigens of MSCs were detected by a flow cytometry [42]. Then, the cells were analyzed by FACS software (Becton Dickinson LSR II, Canton, MA, USA). The eighth passage of the MSCs was used in the current study.

#### 2.3.2. Human Skin Fibroblasts (HSFs)

Human skin fibroblasts were obtained from American Type Culture Collection (ATCC). HSFs can be cultured to higher passages without losing proliferation rate and cell phenotypes, which was described in our previous study [43]. The HSFs were cultured in a low glucose Dulbecco’s Modified Engle’s Medium (DMEM) (Gibco, USA) containing 10% FBS, 1% (*v*/*v*) antibiotics (100 U/mL P/S), and 1% sodium pyruvate (Gibco, Thermo Fisher, Waltham, MA, USA). The cells were harvested using 0.05% trypsin-EDTA after reaching confluency in an incubator (37 °C, 5% CO_2_). The HSFs of the 30th–50th passages were used in the current study.

### 2.4. Biocompatibility Assessments

#### 2.4.1. MTT Assay

The MTT 3-(4,5-dimethylthiazol-2-yl)-2,5-diphenyl tetrazolium bromide solution was used to react with the mitochondrial dehydrogenase in the MSCs and HSFs. After 24, 48, and 96 h of incubation 2 × 10^4^ per well of cells were cultured in a 96-well plate coated with various nanocomposites. The cells were washed after incubation before a 100 μL MTT reagent (0.5 mg/mL) was added to each well for an incubating period of 2 h at 37 °C. Next, a 100 μL dimethylsulfoxide (DMSO) solution was added for 10 min of incubation. The absorbance at 570 nm was read by an ELISA reader (SpectraMax M2, Molecular Devices, San Jose, CA, USA).

#### 2.4.2. Intracellular Reactive Oxygen Species (ROS) Generation

To investigate ROS production, 2 × 10^5^ per well of MSCs and HSFs were cultured with various nanocomposites for 24 and 48 h. The cells were incubated with an oxidation-sensitive fluorescent probe, DCFH-DA (2′,7′-dichlorofluorescin diacetate) (Sigma-Aldrich, Burlington, MA, USA) for 30 min at 37 °C. Next, the ROS were detected by a flow cytometer (BD LSR II, USA) and the fluorescein-positive cells were quantified through FCS software (Becton Dickinson LSR II, Canton, MA, USA).

#### 2.4.3. Examination of Monocyte and Platelet Activities

Human monocytes were acquired from healthy volunteers’ whole blood based on the Percoll protocol (Sigma-Aldrich, Burlington, MA, USA) with IRB approval (CE12164) [44]. The monocytes (1 × 10^5^/well) were seeded in a 24-well plate coated with each nanocomposite and incubated in RPMI medium (10% FBS and 1% (*v*/*v*) antibiotics (10,000 U/mL penicillin G and 10 mg/mL streptomycin)) for 96 h at 37 °C. Next, a 0.05% trypsin solution was used to collect the cells. The cell morphology of the monocytes and macrophages was observed through a microscope. The monocyte conversion yield was calculated according to the following formula: monocyte conversion yield (%) = ((Macrophages)/(Monocytes + Macrophages)) × 100%. To further investigate the inflammatory response, CD68 (marker of macrophages) was also examined through primary anti-CD68 antibody (GeneTex Inc, Irvine, CA, USA) immunofluorescence staining.

To investigate platelet adherence and activation by the different nanomaterials, 2 × 10^6^ cells per well of platelets were seeded for 24 h of incubation. Next, the cells were fixed with a 2.5% glutaraldehyde solution for 8 h. The platelets were then washed with PBS and dehydrated using 30% to 100% alcohol after standing at RT for 10 min. After each sample had been dried out and sputter-coated with gold, the morphology of the platelets affected by the different materials was observed by SEM (JEOL JEM-5200, JEOL Ltd., Akishima, Tokyo, Japan).

#### 2.4.4. Assessments of Cell Apoptosis Induced by Various Nanomaterials

Propidium Iodide (PI) was used to stain nuclei of the MSCs and HSFs (2 × 10^5^ cells/well) after incubating for 24 h, which were then investigated by a flow cytometer. Apoptotic cells were investigated through an Annexin V and PI double staining (Sigma-Aldrich, Burlington, MA, USA) protocol and quantified through a flow cytometer. Indeed, the cells undergoing apoptosis at an early stage tend to expose phosphatidylserine (PS) on the extracellular face of the plasma membrane that can be specifically targeted by Annexin V. Furthermore, PI can permeabilize in the late stage of apoptotic cells to target chromosomes. The dead cells (Annexin V^+^/PI^+^) and apoptosis/necrosis (Annexin V^+^/PI^−^ and Annexin V^−^/PI^+^) cells were detected by a BD FACSCalibur flow cytometer (BD Biosciences, Canada), and the results were analyzed by FlowJo 7.6 software (Becton Dickinson, Canton, MA, USA). All experiments were performed in triplicate.

### 2.5. Investigation of Biological Functions

#### 2.5.1. Cell Migration

First, 96-well plates from a Migration assay kit (Platypus Technologies, Madison, WI, USA) were prepared with the coatings of nanomaterials for 30 min, with the residual solution then removed. Next, the seeding stoppers were cultured with MSCs and HSFs (1 × 10^4^ cells per well) and incubated at 37 °C to reach confluency. Afterwards, the stoppers were removed (with one remaining for pre-migration = 0 h) and others incubated at 37 °C for 24 to 48 h. Next, 200 μL of 2 μM calcein-AM (Sigma-Aldrich, Burlington, MA, USA) was added to each well and stained for 30 min at 0, 24, and 48 h. Ultimately, cell migration ability was observed under a Zeiss Axio Imager A1 fluorescence microscope (White Plains, NY, USA). The boundary moving distance of both the MSCs and HSFs was analyzed and quantified by Image J 5.0 software (National Institutes of Health, Bethesda, MD, USA).

#### 2.5.2. Immunofluorescence Analysis

A total of 1 × 10^4^ cells per well of MSCs were seeded into 24 well plates containing 15 mm coverslip glasses pre-coated with various nanomaterials. After incubation, the cells were washed twice with PBS, and fixed with 4% paraformaldehyde for 30 min at 4 °C. Next, the cells were permeabilized with 0.5% Triton-X 100 (Sigma-Aldrich, Burlington, MA, USA) and blocked with 5% FBS for 30 min at 4 °C. CXCR4 expression was investigated at 48 h and neuronal marker expression was evaluated at day 7. The cells were incubated with several primary antibodies: anti-CXCR4 (1:500 dilution) and anti-nestin, anti-GFAP, and anti-β-tubulin (Santa Cruz, TX, USA) for 8 h at 4 °C, and then washed twice with PBS. Next, we added fluorescein isothiocyanate-conjugated goat anti-mouse/rabbit secondary IgG (1:200 dilution) for a 1 h incubation period at RT. DAPI solution (Invitrogen, White Plains, NY, USA) was used to target cell nuclei. The samples were washed three times at each step, placed on slides using a 50% glycerol/PBS solution, and the slides sealed for use in upcoming observations. The images were captured in a darkroom by a Zeiss Axio Z1 fluorescence microscope (Oberkochen, Germany). The fluorescent positive cells were semi-quantified through Image J software (National Institutes of Health, Bethesda, MD, USA).

#### 2.5.3. Enzyme-Linked Immunosorbent Assay (ELISA)

First, 1 × 10^4^ cells per well of MSCs were cultured in a 12-well plate with the coating of each nanomaterial for 48 h. Next, after being centrifuged, the supernatants were collected and the cytokine, SDF-1α, had its expression level determined by an ELISA kit (R&D, Minneapolis, MN, USA) following the manufacturer’s instructions. The results were analyzed through an enzyme-linked immunosorbent assay (ELISA) reader (SpectraMax M2, Molecular Devices, San Jose, CA, USA). The data were obtained from quadruplicate experiments.

#### 2.5.4. Assessment of Matrix Metalloproteinase (MMP) Activities

A total of 2 × 10^5^ cells per well of MSCs and HSFs were cultured in 6-well plates with the coatings of various nanocomposites. Next, after 48 h of incubation, the cultured medium in each well was collected to process the gelatin zymography assay. The protease-digested area exhibits itself as clear bands on a dark blue background. Next, the gel was digitized by a densitometer, with the MMP digested bands semi-quantified by Image Pro Plus 5.0 software (Media Cybernetics, Burlington, MA, USA).

#### 2.5.5. Real-Time PCR assay

The total amount of mRNA in the MSCs was extracted based on the procedure provided by the manufacturer. First, 1 × 10^5^ per well of MSCs were seeded in 10 cm culture dishes with the coatings of various nanocomposites and then incubated for 3, 5, and 7 days. Next, the cells were treated with 1 mL of TRIzol (Invitrogen, Thermo Fisher, Waltham, MA, USA) for 5 min and 200 μL of chloroform (Sigma-Aldrich, Burlington, MA, USA) for 15 s to extract RNA, and were finally allowed to stand for 3 min at RT. Furthermore, the cells were centrifuged at 12,000 rpm/4 °C for 15 min. Next, the supernatant was removed and 500 μL of 4 °C isopropanol was then added to incubate for 15 min. Each sample was centrifuged at 12,000 rpm/4 °C for 15 min. Next, the supernatant was removed and washed twice with 1 mL of 75% alcohol. After drying the RNA samples, 20 μL of DEPC-treated H_2_O soluble precipitate was added into each sample, and quantified by the absorbance at 260 nm using a SpectraMax M2 ELISA reader (Molecular Devices, San Jose, CA, USA).

A RevertAid™ First Strand cDNA Synthesis Kit (Thermo Fisher, Pittsfield, MA, USA) was used for the synthesis of cDNA. Then, 2 μL of oligo (dT) 18 primers and random hexamers (1:1) were added to each RNA sample, with the samples then placed into a gradient polymerase reaction temperature controller at 65 °C for 5 min. Before the RNA samples were reacted at 42 °C for 60 min, 4 μL of 5× reaction buffer, 1 μL of LockTM RNase inhibitor (20 U/mL), 2 μL of dNTP Mix (10 mM), and 1 μL of RevertAid™ M-MuLV Reverse Transcriptase (200 U/mL) were all added to each sample. Finally, the samples were carried out to obtain the cDNA at 70 °C for 5 min.

The polymerase chain reaction (PCR) was processed by the 1Q2 Fast qPCR System using the cDNA as a template with a 10 μL reaction volume according to the manufacturer’s procedures. A total of 0.5 μL of primer (0.3 μM) and 5 μL of enzyme were added to the cDNA sample, and the mRNA expression was subsequently analyzed by the StepOnePlus™ Real-Time PCR System.

### 2.6. Rat Subcutaneous In Vivo Model

Two- to three- month-old female Sprague Dawley (SD) rats weighing 300–350 g were obtained from the National Laboratory Animal Center, Taiwan. The experimental procedures were processed with approval from the Animal Care and Use Committee (La-1,071,565). In this study, gold nanoparticle solution was coated on a glass coverslip (15 mm) and implanted to rat subcutaneous tissue. It can be calculated that the volume of each gold nanoparticle (AuNP) (~5 nm) is 1.9635 × 10^−13^/cm^2^. The density of gold is 19,320 mg/cm^3^. It can be calculated that the weight of a AuNP is 3.79348 × 10^−9^ mg. For coating on a glass coverslip (15 mm), 552 μL of 50 ppm AuNP solution was added on the glass coverslip. It can be calculated that the solution contained 0.0028 mg of AuNP, which can be converted into 72,787 AuNPs on the coating. After being given local anesthesia, the dorsal skin of each rat was cautiously incised by 10 mm^2^ in order to implant various materials. Next, the wound tissue was resected for investigation after a 1-month implantation. A fibrous capsule formation in six sites was investigated by hematoxylin and eosin (H&E) staining, and the average encapsulated fibrotic tissues quantified by commercial software. Masson’s trichrome staining kits were used to detect collagen deposition in tissue based on the manufacturer’s instructions (Sigma-Aldrich, Burlington, MA, USA). The areas of fibrosis tissue and collagen deposition were measured using Image J 5.0 software (National Institutes of Health, Bethesda, MD, USA). To investigate endothelialization in the tissue area influenced by each nanomaterial, mouse monoclonal anti-CD31 antibodies were used. To measure leukocyte infiltration, APC anti-mouse CD45 antibodies were used. To investigate M2 polarization of macrophages, mouse monoclonal anti-CD163 antibodies (Santa Cruz, Dallas, TX, USA) in a 1:200 dilution were used. Donkey anti-mouse IgG secondary antibodies (AF488, Invitrogen, White Plains, NY, USA) and anti-mouse immunoglobulin G (Jackson ImmunoResearch, West Grove, PA, USA) in a 1:500 dilution were both used for signal amplification. To evaluate M1 polarization, the tissue area implanted with each nanomaterial was first incubated with mouse monoclonal anti-CD86 antibodies (Santa Cruz, Dallas, TX, USA) in a 1:200 solution, and further incubated with second antibodies. Next, the positive staining was visualized with DAB (3,3′-diaminobenzidine) and counter stained with hematoxylin. Furthermore, the apoptotic cells in the rat model induced by the various nanocomposites were detected by TUNEL assay. DNA fragmentations could be targeted by labeling 3′-hydroxyl termini during the late stage of cell apoptosis. An In Situ Cell Death Detection Kit, AP (Roche Diagnostics, Indianapolis, IN, USA) was purchased to target the apoptotic cells based on the manufacturer’s protocol. An Olympus IX71 fluorescence microscope (Tokyo, Japan) was used to determine the fluorescent intensity. Cell nuclei were targeted by a DAPI solution. The number of rats was 5 (n = 5). Data were represented as mean ± SD.

### 2.7. Statistical Analysis

All the experiments were independently processed three times in order to avoid uncertainty, with the results represented as mean ± standard deviation (SD). Student’s *t*-test and the SPSS Statistics v17.0 method were applied to evaluate the statistical difference between groups. A *p* value less than 0.05 was considered statistically significant.

## 3. Results

### 3.1. Characterization of Pul-Derived Nanocomposites

Figure 1A shows the SEM image of Pul–Col–Au nanoparticles. The size of Pul–Col–Au was 3.8 ± 0.2 nm and larger than gold nanoparticle (3–5 nm). The above evidence indicated that the pullulan and collagen molecular chains were entangled around the gold nanoparticle. Further, the Pul–Col–Au nanoparticles were observed by TEM assay (Figure 1B). The zeta potential of Pul–Col–Au was also investigated by DLS assay (Figure 1C). The water contact angles of pure Pul, pure Col, Pul–Col, Pul–Au, Pul–Col–Au, and Col–Au were 100.17°, 79.17°, 86.97°, 89.33°, 74.87°, and 61.9°, respectively. The results indicate that the addition of Au nanoparticles led to the surface of Pul–Col–Au becoming more hydrophilic, thus promoting the cell adhesion ability of Pul–Col–Au nanocomposites (Figure 1D). To identify the decoration of Au, the absorption wavelength of each material was measured through the use of UV-Vis spectroscopy. Figure 1E indicates that a typical peak at 520 nm for Au nanoparticle decoration was observed in the Pul–Au, Col–Au, and Pul–Col–Au groups.

Afterwards, the FTIR assay was processed to investigate the specific functional groups of each nanomaterial, with the results shown in Figure 2A. The spectra of pure Pul indicated the specific peaks to be at 929 cm^−1^ (C–O–C bond), 1029 cm^−1^ (C–O vibration), and 2931 cm^−1^ (CH_2_/CH_3_). The pure Col was also measured, with the peaks located at 1540 cm^−1^, 1655 cm^−1^, and 3200 cm^−1^ being attributed to the vibration bands of amide II, amide I, and the O–H bond, respectively. After Pul was combined with Col, the absorption bands of the C–O–C vibration shifted from 929 cm^−1^ to 910 cm^−1^, C–O vibration shifted from 1029 cm^−1^ to 1033 cm^−1^, while the CH_2_/CH_3_ functional group was observed to shift from 2931 cm^−1^ to 2935 cm^−1^, and the amide II band shifted from 1540 cm^−1^ to 1550 cm^−1^. Moreover, N–H stretching at 3340 cm^−1^ was found in the Pul–Col–Au group. The above evidence validates that Au nanoparticles successfully bonded with Pul and Col. 

The functional groups of different nanocomposites were also examined through SERS, with the results shown in Figure 2B. The specific peaks of pure Pul were located at 1260 cm^−1^, 1342 cm^−1^, and 1474 cm^−1^, and represented as COH, CH_2_/CH_3_, and CH_2_/CH_3_, respectively. The specific peaks of pure Col were 1255 cm^−1^ (amide III), 1454 cm^−1^ (CH_2_/CH_3_), and 1688 cm^−1^ (amide I). After Au was combined with Pul, the peaks were displayed as 1279 cm^−1^ (COH), 1355 cm^−1^ (CH_2_/CH_3_), 1466 cm^−1^ (CH_2_/CH_3_), and 1605 cm^−1^ (C=C). In the Pul–Col group, the absorption peaks were indicated as 1263 cm^−1^ (amide III), 1347 cm^−1^ (CH_2_/CH_3_), 1470 cm^−1^ (CH_2_/CH_3_), and 1688 cm^−1^ (amide I). Furthermore, the specific absorption peaks of Pul–Col–Au nanocomposites were located at 1259 cm^−1^ (amide III), 1347 cm^−1^ (CH_2_/CH_3_), 1470 cm^−1^ (CH_2_/CH_3_), and 1688 cm^−1^ (amide I). The above results elucidate that the signals of Pul and Col could be clearly observed in the Pul–Au, Pul–Col, and Pul–Col–Au groups.

The various nanocomposites were also identified by XPS, with the results displayed as Figure 2C. Due to pure Pul not containing N, there was no N signaling. After Pul was combined with Col, the N1s peak position at 402.6 eV binding energy was observed owing to the amide group in Col. Furthermore, after Au nanoparticles were decorated onto Pul–Col, the Au peak position was measured at 86 and 91.8 eV binding energy, which was represented as Au4f_7/2_ and Au4f_5/2_, respectively. The above evidence reveals that the addition of Au nanoparticles did not cause the structural changing of Pul and Col.

### 3.2. Biocompatibility Assessments of Pul-Derived Nanocomposites for MSCs and HSFs

The specific surface markers of MSCs were firstly detected by a flow cytometry. The markers, CD34 and CD45, were highly expressed in endothelial cells and immune cells, which were represented as negative markers (0.87% and 0.8%, respectively) (Appendix A). The specific positive markers CD44 (99%) and CD90 (99.6%) for MSCs were remarkably detected (Appendix A). Then, the MSCs were applied in the following experiments.

Next, we investigated the cytotoxicity of Pul–Col–Au, Pul–Au, Pul–Col, and pure Pul to confirm the optimal concentration of Pul on MSCs and HSFs. In Appendix A, after culturing MSCs with Pul–Col–Au 0.125, 0.25, 0.5, and 1 g/mL, the cell viability was 0.51, 0.66, 0.43, and 0.47 at 24 h compared to the control, while at 48 h it was 1.15, 1.16, 1.06, and 0.95 compared to the control, respectively. After culturing MSCs with Pul–Au 0.125, 0.25, 0.5, and 1 g/mL, the cell viability at 24 h was 0.56, 0.57, 0.51, and 0.56, respectively, while at 48 h it was 0.96, 1.23, 1.23, and 1.18 compared to the control. After culturing MSCs with Pul–Col 0.125, 0.25, 0.5, and 1 g/mL, the cell viability at 24 h was 0.71, 0.77, 0.74, and 0.64, while at 48 h it was 0.90, 1.0, 0.91, and 0.86 compared to the control, respectively. After culturing MSCs with pure Pul 0.125, 0.25, 0.5, and 1 g/mL, the cell viability at 24 h was 0.69, 0.83, 0.76, and 0.79, while at 48 h it was 0.83, 1.07, 0.85, and 0.81 compared to the control, respectively. The above results demonstrate that the concentration of Pul at 0.25 g/mL was the appropriate amount for MSC proliferation. Next, HSFs were also cultured with Pul–Col–Au, Pul–Au, Pul–Col, and pure Pul at the same condition (0.125, 0.25, 0.5, and 1 g/mL) to investigate cytotoxicity (Appendix A). In the Pul–Col–Au groups, the cell viability at 24 h was 0.55, 0.57, 0.47, and 0.44, while at 48 h it was 0.91, 0.98, 0.88, and 0.80 compared to the control, respectively. In the Pul–Au groups, the cell viability at 24 h was 0.56, 0.60, 0.59, and 0.56, while at 48 h it was 0.89, 1.01, 0.87, and 0.80 compared to the control, respectively. In the Pul–Col groups, the cell viability at 24 h was 0.70, 0.75, 0.66, and 0.65, while at 48 h it was 0.81, 0.85, 0.82, and 0.80 compared to the control, respectively. In the pure Pul groups, the cell viability at 24 h was 0.65, 0.70, 0.68, and 0.68, while at 48 h it was 0.81, 0.82, 0.80, and 0.80 compared to the control, respectively. The above data also indicate that the concentration of Pul at 0.25 g/mL was the appropriate concentration for HSF proliferation.

After having confirmed the concentration of Pul at 0.25 g, MSCs and HSFs were cultured on various nanocomposites to evaluate cell proliferation. The results indicate that the cell viability of MSCs significantly increased in the Pul–Col–Au group (OD_570 nm_ = 1.24) when compared to other groups at 24 h (MSC, Pul, Col, Pul–Au, Pul–Col, and Col–Au: OD_570 nm_ = 0.93, 0.94, 1.03, 1.14, 1.03, and 1.15, respectively) (Figure 3A, left panel). A similar trend also occurred at 48 h, when cell viability was 0.95, 1.05, 1.26, 1.19, 1.48, and 1.41 (Pul, Col, Pul–Au, Pul–Col, Pul–Col–Au, and Col–Au, respectively) (Figure 3A, right panel). The cell viability of HSFs seeding on different materials was also investigated. Figure 3B, left panel demonstrated that the cell viability of HSFs remarkably increased in the Pul–Col–Au group (OD_570 nm_ = 1.37) when compared to other groups at 24 h (Pul, Col, Pul–Au, Pul–Col, and Col–Au: OD_570 nm_ = 1.09, 1.19, 1.21, 1.12, and 1.23, respectively). Moreover, as seen in Figure 3B, right panel, the cell viability of HSFs at 48 h in the Pul–Col–Au groups (OD_570n_ = 1.47) was significantly greater than that of the other groups (Pul, Col, Pul–Au, Pul–Col, and Col–Au: OD_570_ = 1.28, 1.29, 1.37, 1.31, and 1.41, respectively). Additionally, the cell viability in the long-term culture (96 h) for both MSCs and HSFs was examined. The results for MSCs at 96 h were significantly higher in Col–Au, Pul–Col–Au, and Pul–Au groups (OD_570 nm_ = 1.84, 1.8, 1.76, respectively), followed by Pul–Col, MSC alone, pure Pul, and pure Col groups (OD_570 nm_ = 1.64, 1.62, 1.6, 1.58, respectively) (Appendix A). The data for HSFs at 96 h were remarkably greater in the Pul–Au and Pul–Col–Au groups, and Col–Au (OD_570 nm_ = 1.86, 1.85, 1.84, respectively), followed by pure Col, HSF alone, Pul–Col, and pure Pul groups (OD_570 nm_ = 1.8, 1.79, 1.78, 1.76, respectively) (Appendix A). The evidence indicated that the cell viability of the Pul–Col–Au group was significantly higher in both MSCs and HSFs. 

The intracellular ROS production of HSFs and MSCs culturing with different materials was also investigated at 24 and 48 h. The semi-quantitative data at 24 h indicated that the average amount of ROS production in HSFs in the Pul–Col–Au group (~0.29-fold) was remarkably less than that of the other groups when compared to the control (Pul, Col, Pul–Au, Pul–Col, Col–Au: ~0.92-fold, ~0.88-fold, ~0.60-fold, ~0.89-fold, ~0.56-fold) (Figure 3C, left panel). Also, in MSCs at 24 h, ROS generation in the Pul–Col–Au group (~0.37-fold) was significantly lower than that of the other groups when compared to the control (Pul, Col, Pul–Au, Pul–Col, Col–Au: ~0.76-fold, ~0.74-fold, ~0.60-fold, ~0.61-fold, ~0.59-fold) (Figure 3C, right panel). Additionally, the quantitative data at 48 h are demonstrated in Figure 3D. In HSFs, the ROS production in the Pul–Col–Au group was ~0.28-fold, followed by the Col–Au group (~0.36-fold), Pul–Col group (~0.51-fold), Pul–Au group (~0.57-fold), Col group (~0.58-fold), and Pul group (~0.65-fold) (Figure 3D, left panel). In MSCs, the ROS production in the Pul–Col–Au group was ~0.55-fold, followed by the Col–Au group (~0.64-fold), Pul–Au group (~0.75-fold), Pul–Col group (~0.79-fold), Col group (~0.83-fold), and Pul group (~0.87-fold) (Figure 3D, right panel). The above evidence elucidates that the Pul–Col–Au nanocomposite exhibits superior anti-ROS generation ability.

The expression of macrophage marker CD68 was demonstrated by immunofluorescence staining (Figure 4A) and the semi-quantitative result based on fluorescent intensity showed the lowest expression to be in the Pul–Col–Au group (~0.51-fold), followed by the Pul–Au group (~0.62-fold), Col–Au group (~0.69-fold), Pul–Col group (~0.76-fold), Col group (~0.78-fold), and Pul group (~0.94-fold) when compared to the control (Figure 4C). Furthermore, platelets and monocytes will rapidly activate when inflammatory response occurs. The degree of platelet activation from various materials as observed by SEM is shown in Figure 4B. In the Pul–Col–Au group, the platelets were mostly represented as round morphology (non-activated form), while in the pure Pul, pure Col, and Pul–Col groups the platelets were subsequently exhibited as being flattened (active form). The semi-quantitative data for platelet adherence were lowest in the Pul–Col–Au group (~0.51-fold), followed by the Pul–Au group (~0.56-fold), Col–Au group (~0.69-fold), Pul–Col group (~0.86-fold), pure Col group (~0.9-fold), and pure Pul group (~0.99-fold) (Figure 4D). Moreover, the semi-quantitative results for platelet activation were the lowest in the Pul–Col–Au (~0.6-fold) and Col–Au groups (~0.58-fold), followed by the Pul–Au group (~0.65-fold), Pul–Col group (~0.86-fold), pure Col group (~0.92-fold), and pure Pul group (~0.94-fold) (Figure 4E).

The ratio of monocytes (~5 µm) transformed to macrophages (~40 to 45 µm) on different materials after 96 h of incubation is shown in Appendix A. Both the number of monocytes and the number of macrophages w lowest in the Pul–Col–Au group (Appendix A). The monocyte conversion yield was also lowest in the Pul–Col–Au group (~0.51-fold), followed by the Pul–Au group (~0.62-fold), Col–Au group (~0.69-fold), Pul–Col group (~0.76-fold), pure Col group (~0.78-fold), and pure Pul group (~0.94-fold) (Appendix A). The above evidence demonstrates that the Pul–Col–Au nanocomposite exhibited superior anti-inflammatory abilities to maintain better biocompatibility.

### 3.3. Cell Apoptosis Induced by Pul-Derived Nanocomposites

Cell survival and apoptosis influenced by each nanomaterial for MSC and HSF at 24 h was investigated by Annexin V/PI double staining assay and detected by flow cytometry (Figure 5A,B). As presented in Figure 5C, the viable cell populations of MSCs and HSFs demonstrate no significant difference when compared to the control. Moreover, the results seen in Figure 5D indicate that the apoptotic cell population of MSCs was lowest in both the Pul–Col–Au (~0.2-fold) and Col–Au (~0.2-fold) groups, followed by the Pul–Au group (~0.22-fold), Pul–Col group (~0.3-fold), pure Col group (~0.4-fold), pure Pul group (~0.4-fold), and MSC alone group (1-fold). The apoptotic population of HSFs was also the lowest in the Pul–Col–Au group (~0.14-fold), followed by the Col–Au group (~0.21-fold), Pul–Au group (~0.28-fold), Pul–Col group (~0.43-fold), pure Col group (~0.48-fold), pure Pul group (~0.51-fold), and HSF alone group (1-fold). The above results strongly indicate that each nanomaterial would not lead to cell apoptosis in MSCs and HSFs, particularly in the Pul–Col–Au group, thus demonstrating that the Pul–Col–Au nanocomposite is a better material for biocompatibility.

### 3.4. Cell Migration Affected by Pul-Derived Nanocomposites

The SDF-1α/CXCR4 pathway as well as metalloproteinase (MMP) activity play important roles in cell migration for tissue regeneration. The immunofluorescence (IF) images for CXCR4 expression in MSCs at 48 h are displayed in Figure 6A. The expression based on IF intensity demonstrated that the greatest expression of CXCR4 was seen in Pul–Col–Au (~1.82-fold), followed by Col–Au (~1.71-fold), Pul–Au (~1.54-fold), Pul–Col (~1.24-fold), pure Col (~1.21-fold), and pure Pul (~1.14-fold) (Figure 6B). Furthermore, the expression of CXCR4 investigated by the FACS method is demonstrated in Figure 6C. The results indicate a better expression in the Pul–Col–Au (~1.4-fold) and Col–Au (~1.44-fold) groups, followed by Pul–Au (~1.38-fold), Pul–Col (~1.28-fold), pure Col (~1.26-fold), and pure Pul (~1.18-fold). The flow gating results were also demonstrated in Appendix A to support the above FACS data. Additionally, the expression of SDF-1α secreted from MSCs culturing with various nanocomposites was measured through ELISA assay after 48 h of incubation. The amount of SDF-1α secreted from MSCs in the Pul–Col–Au group (~1.45-fold) was greater than that of the other groups (pure Pul group (~1.06-fold), pure Col group (~1.11-fold), Pul–Au group (~1.25-fold), Pul–Col group (~1.13-fold), and Col–Au group (~1.34-fold)) when compared to the control (Figure 6D). The real time images of MSC migration after culturing with various materials were evaluated by calcein-AM staining at both 24 and 48 h (Figure 6E). The boundary moving distance of MSCs at 24 h in the pure Pul, pure Col, Pul–Au, Pul–Col, Pul–Col–Au, and Col–Au groups was 36.05 μm, 38.01 μm, 37.15 μm, 38.5 μm, 52.57 μm, and 47.3 μm, respectively. Also, at 48 h, the quantitative results of the groups were 51.04 μm, 53.59 μm, 70.58 μm, 66.86 μm, 72.09 μm, and 65.35 μm, respectively (Figure 6F). MMP activity and MSCs induced by different materials were also investigated (Figure 6G). The expression of MMP-2 from MSCs at 48 h in the pure Pul, pure Col, Pul–Au, Pul–Col, Pul–Col–Au, and Col–Au groups was ~1.39-fold, ~1.86-fold, ~2.07-fold, ~1.63-fold, ~2.65-fold, and ~2.25-fold, respectively. Additionally, the expression of MMP-9 from MSCs at 48 h in the pure Pul, pure Col, Pul–Au, Pul–Col, Pul–Col–Au, and Col–Au groups was ~1.19-fold, ~1.25-fold, ~1.32-fold, ~1.21-fold, ~1.43-fold, and ~1.27-fold, respectively (Figure 6H).

Furthermore, the cell migration of HSFs was also investigated, with the images displayed in Appendix A. At 24 h, the moving distance in the pure Pul, pure Col, Pul–Au, Pul–Col, Pul–Col–Au, and Col–Au groups was 32.02 μm, 32.12 μm, 42.02 μm, 29.88 μm, 45.78 μm, and 43.61 μm, respectively. At 48 h, the quantified results of the groups were 47.91 μm, 52.73 μm, 63.93 μm, 51.44 μm, 69.06 μm, and 64.59 μm, respectively (Appendix A). The zymogram image of MMP-2/9 expression in HSFs is also shown in Appendix A. The expression of MMP-2 and MMP-9 in HSF seeding on various materials was also examined. The expression of MMP-2 in HSFs at 48 h in the Pul, Col, Pul–Au, Pul–Col, Pul–Col–Au, and Col–Au groups was ~1.03-fold, ~1.21-fold, ~1.35-fold, ~1.2-fold, ~175-fold, and ~1.59-fold, respectively. Additionally, the expression of MMP-9 from HSFs at 48 h in the Pul, Col, Pul–Au, Pul–Col, Pul–Col–Au, and Col–Au groups was ~1.13-fold, ~1.36-fold, ~1.92-fold, ~1.37-fold, ~2.14-fold, and ~2.02-fold, respectively (Appendix A). The above findings suggest that Pul–Col–Au nanocomposites could promote the expression of SDF-1α/CXCR4 and further facilitate MMP-2/9 activities to enhance MSCs migration. 

### 3.5. Neuronal Differentiation Capacity Induced by Pul–Col–Au Nanocomposites

The neuronal differentiation capacities of MSCs stimulated by various nanocomposites was evaluated. The markers of neurons, nestin, GFAP, and β-tubulin were all observed at day 7, with the immunofluorescent images displayed in Figure 7A–C, respectively. Based on the semi-quantitative results, Pul–Col–Au induced the highest expression level of neural markers in MSCs at day 7 when compared to the control (Figure 7D,F). The semi-quantitative data based on fluorescent intensity demonstrated ~2.43-fold for nestin (Figure 7D), ~3.83-fold for GFAP (Figure 7E), and ~1.52-fold for β-tubulin (Figure 7F) at day 7. Furthermore, we evaluated the mRNA expression of nestin, GFAP, and β-tubulin through real-time PCR assay at days 3, 5, and 7. The semi-quantitative results are demonstrated in Appendix A. At day 3, the expression of nestin, GFAP, and β-tubulin was better in the Pul–Col–Au group, with the results represented as ~3.01-fold, ~3.36-fold, and ~2.46-fold, respectively. At day 5, Pul–Col–Au induced the highest expression of neuronal markers in MSCs, with the results being ~5.89-fold, ~8.22-fold, and ~9.23-fold, respectively. Finally, at day 7, the expression of nestin, GFAP, and β-tubulin promoted by Pul–Col–Au was also the greatest when compared to the control. The results were semi-quantified as ~6.09-fold, ~7.86-fold, and ~6.54-fold, respectively. The above evidence elucidates that Pul–Col–Au can significantly stimulate neuronal differentiation in MSCs.

### 3.6. Subcutaneous Implantation for Pul–Col–Au Nanocomposites

After subcutaneous implantation of various materials into our rat models for 4 weeks, the biocompatibility effects and inflammatory responses were further investigated. Leukocytes infiltration would be induced by the M1 polarization macrophages [45]. The leukocyte infiltration in animal tissue was detected through CD45 expression, with the images demonstrated in Figure 8A. The semi-quantitative results indicate that the lowest expression was seen in the Pul–Col–Au group (~0.56-fold), followed by the Col–Au (~0.64-fold) and Pul–Au (~0.7-fold) groups (Figure 8D). M2 polarization macrophages contributed to the inhibition of inflammation and the strengthening of angiogenesis [46], which were evaluated by CD163 IHC staining (Figure 8B). The quantitative results show that the highest expression was seen in the Pul–Col–Au (~1.45-fold) group, followed by the Col–Au (~1.41-fold) and Pul–Au (~1.25-fold) groups (Figure 8E). Furthermore, the CD86 marker of M1 polarization macrophages induced the pro-inflammatory cytokines expression [46], which was investigated by DAB staining (Figure 8C). Based on those semi-quantitative results, the Pul–Col–Au (~0.48-fold) group induced the lowest expression of CD86, follow by the Col–Au (~0.5-fold) and Pul–Au (~0.62-fold) groups when compared to the control (Figure 8F).

Endothelialization capacity was measured by CD31 expression after implantation. The images are displayed in Figure 9A. The results have been semi-quantified in Figure 9E and demonstrate the better expression seen in the Pul–Col–Au (~1.24-fold), Col–Au (~1.26-fold), and Pul–Au (~1.28-fold) groups when compared to the control. The fibrous capsule formation was measured by H&E staining (Figure 9B), with the semi-quantitative results indicating that the lowest formation occurred in the Pul–Col–Au group (~0.56-fold), followed by the Col–Au (~0.64-fold) and Pul–Au (~0.7-fold) groups (Figure 9F). Collagen deposition was examined through Masson’s trichrome staining (Figure 9C), where the results also indicated that the deposition was the lowest in the Pul–Col–Au group (~0.36-fold), followed by the Col–Au (~0.38-fold) and Pul–Au (~0.62-fold) groups (Figure 9G). Additionally, for biosafety assessment, the TUNEL assay was applied to target the late stage of apoptotic cells (DNA fragmentation), with the images demonstrated in Figure 9D. The number of TUNEL-positive cells was the lowest in the Pul–Col–Au group (~0.8-fold), followed by the Pul–Au (~0.82-fold) and Col–Au (~0.87-fold) groups when compared to the control (Figure 9H). The current research has elucidated that Pul–Col–Au nanocomposites could demonstrate better biocompatibility, biological performance, and superior neuronal differentiation capacity. In vivo assay also indicated that Pul–Col–Au exhibited the advanced abilities of anti-immune response and more bio-safety.

## 4. Discussion

Nanoparticles have been investigated and developed for various applications in biomedicine. Due to their small size, they more easily interact with cells to influence cell behavior [47]. A previous study investigated both the cytotoxicity and immunogenic responses of Au nanoparticles on RAW 264.7 macrophage cells. The results of the study demonstrated that there was no cytotoxicity due to the reduction of ROS species [48]. Furthermore, the surface modification caused by Au nanoparticles has attracted interest. The amine and thiol groups binding of gold nanoparticles enabled the modification through proteins and polymers [49] that could be applied to clinical applications such as drug delivery and tissue regeneration engineering [50]. For example, Au nanoparticles could assemble with polyethylene glycol by the thiol group bond [51]. Furthermore, the surface modified nanocomposites containing Au nanoparticles have been verified to decrease platelet activation and monocyte conversion, as well as to enhance cell proliferation, anti-oxidative abilities, and anti-inflammatory capacities [52]. Meanwhile, a study found in the literature concluded that Au nanoparticles at the concentration of 12.2 ppm promote cell proliferation and migration [52]. Additionally, Au nanoparticles were verified to have anti-oxidative abilities [27]. We first prepared the combination of Col and Pul at the concentration of 0.25 g/mL and Au nanoparticles at 12.2 ppm. In line with our findings, after Pul–Col was incorporated with Au nanoparticles, cell viability significantly increased, while ROS production decreased for both MSCs and HSFs. Moreover, performances of anti-immune response were also demonstrated in Pul–Col–Au (Figure 3 and Figure 4).

Tissue repair can be improved by angiogenic cytokines such as the basic fibroblast growth factor (bFGF), stromal-derived factor-1α (SDF-1α), and vascular endothelial growth factor (VEGF), which are all associated with tissue repair [42]. Thus, VEGF and SDF-1α can regulate and improve the vascular tissue repair process. For instance, interactions between SDF-1α and CXCR4 can associate the mobilization of MSCs into ischemic tissue [42]. The mobilization of MSCs subsequently secrete VEGF and SDF-1α factors in tissues, which can induce the formation of blood vessels [53]. Additionally, matrix metalloproteinase (MMP-2/9) activities also regulated cell migration and differentiation ability [54]. In line with our results, Pul–Col–Au nanocomposites induced the expression of the CXCR4/ SDF-1α axis, and matrix metalloproteinase (MMP-2/9) activities for MSC and HSF migration (Figure 6 and Appendix A).

HSFs have been well applied for skin repair applications. Previous literature described a cellularized bilayer skin substitute comprising of pullulan–gelatin hydrogel (PG-1), primary HSFs, and keratinocytes. PG-1 provided a biocompatible environment for the bilayer created by HSFs and keratinocytes, and further promoted the formation of thicker neodermis [33]. Furthermore, researchers had created Pul–Col composite hydrogels to investigate in vitro biocompatibility. The human foreskin fibroblasts exhibited significant viability (>97%) after culturing with Pul–Col hydrogels and demonstrated better invasion and attachment [37]. The above evidence shows that the combination of Pul-based nanocomposites and HSFs has a strong potential for wound healing (Figure 3B and Appendix A). MSCs are an attractive cell source for tissue repair, owing to their capacity for differentiation and the secretion of bioactive factors which are immunomodulatory for therapeutic applications [55]. MSCs can be discovered in various locations such as bone marrow, the umbilical cord, and adipose tissue, as well as others [56,57,58]. Furthermore, the literature has reported that rat nestin-positive MSCs promoted an astrocytic cell fate in mouse embryonic neural progenitor cells, and also expressed the astrocytic marker, GFAP [59]. Based on the current research, MSCs have been cultured with Pul–Col–Au to investigate their neuronal differentiation capacity. Nestin is a type VI intermediate filament protein, which is the protein marker in neurons [60]. Astroglial cells are present in the central nervous system of mammalian animals, and highly express glial fibrillary acidic protein (GFAP) [61]. β-Tubulin is considered to be an early marker for neuronal differentiation and is also expressed in carcinoma cells [62]. Our results demonstrate that the expression of nestin, GFAP, and β-tubulin is significantly induced by Pul–Col–Au, indicating the potential for neuronal differentiation in MSCs.

Poly(lactide-co-glycolide) scaffolds containing β-glucans are highly biocompatible for human fibroblasts and adipose-derived stem cells [63]. Also, gelatin/glucan scaffold culturing with keratinocytes and fibroblasts can improve the wound healing process and also indicate the potential for glucan nanomaterials to be used in skin repair [64]. Additionally, the regulation of macrophage polarization is efficient for tissue regeneration engineering. Macrophage conversion from inflammatory M1 to fibrotic M2 polarization may help in diseases such as chronic wounds which are in an inflammatory state [65]. While inflammatory responses do occur, M1 phenotype macrophages induce the expression of pro-inflammatory cytokines such as TNF-α and IL-1β. Neutrophils infiltration would also occur after M1 macrophage polarization. In contrast, M2 fibrotic phenotype macrophages inhibited inflammatory response and enhanced angiogenesis and matrix deposition [66]. A report in the literature also demonstrated the M1 to M2 polarization of macrophages by using retinoic acid with MSCs culturing on an electrospun pullulan/gelatin scaffold. These results confirmed that the combination of retinoic acid and MSCs would convert M1 polarization to become an M2 macrophage [67]. In line with our in vivo results, we chose CD86 as our M1 phenotype, and CD 163 as our M2 phenotype [46], with Pul–Col–Au nanocomposites significantly decreasing the amount of M1 macrophages, but increasing the amount of M2. Moreover, the antigen CD45 was used to target leukocyte infiltration [45], with the results being the lowest after implanting Pul–Col–Au in our rat models. Moreover, a study used acetylated polysaccharide and pullulan acetate combined with vascular endothelial growth factor (VEGF) to investigate endothelization. The results confirmed that pullulan acetate with VEGF could stimulate the endothelialization of human umbilical vein endothelial cells [68]. Our results also further demonstrate that Pul–Col–Au enhanced endothelialization capacity in vivo.

Several studies have described Pul-based nanocomposites for promoting wound healing. Aminoalkysilane was grafted on a bacterial nanocellulose (BNC) membrane to form A-*g*-BNC, then combined with fabricated pullulan–zinc oxide (Pul–ZnO) to generate A-*g*-BNC/Pul–ZnO nanoparticles. A-*g*-BNC/Pul–ZnO exhibited a better biocompatibility for L929 fibroblast cells, and the animal assessments elucidated the enhancement of re-epithelialization, the formation of blood vessels, which is associated with wound repair [69]. Another study verified that a curcumin grafted hyaluronic acid modified pullulan polymer (Cur–HA–SPul) could facilitate L929 cells proliferation and anti-microbial capacity. The rat model results also exhibited the promotion of wound healing [70]. A novel three-dimensional film, chitosan/carboxymethyl pullulan polyelectrolyte complex (PEC) loaded with 45S5 bioglass (CCMPBG), was reported to enhance mechanical strength and biodegradation behavior for wound tissue regeneration [71]. Additionally, a research team developed a hydrogel comprising Pul–Col nanocomposites and adipose-derived stem cells (ASC), and the hydrogel induced better re-epithelialization, vascularization, and expression of the angiogenesis-related genes VEGF and SDF-1α [72].

Our recent studies also demonstrated various nanocomposites for tissue regeneration. The component of the extracellular matrix, fibronectin (FN), was fabricated with silver nanoparticles (AgNPs). AgNPs were verified to have an anti-bacterial ability. In vitro and in vivo measurements showed that FN–AgNPs significantly inhibited the expression of pro-inflammatory cytokines (TNF-α, IL-1β, and IL-6), promoted endothelialization for MSCs, and attenuated foreign body responses, demonstrating the potential vascular repair applications of FN–AgNPs [73]. Graphene oxide decorated with Au nanoparticles (Go–Au) was suggested to be a promising nanocarrier owing to its better biocompatibility and its ability to induce various differentiated cell types for MSCs such as neuron, endothelial cells, osteocytes, and adipocytes [74]. The synthetic polymer, polyethylene glycol (PEG), was combined with Au nanoparticles. The results indicated the surface modification of Au nanoparticles on the PEG film significantly improved biological performance and biocompatibility [75]. The above evidence indicates that nanoparticles such as gold and silver could enhance the functions of biomaterials. In accordance with the present study, Pul–Col–Au exhibited a better biocompatibility and significantly induced the neuronal differentiation of nestin, β-tubulin, and GFAP for MSCs. In vivo assessments also confirmed Pul–Col–Au nanocomposites to be safe materials for implantation. The above evidence suggest that Pul–Col–Au nanocomposites are an efficient source for clinical wound healing applications.

## 5. Conclusions

In the present research, we prepared nanocomposites by combining them with pullulan, collagen, and Au nanocomposites (Pul–Col–Au) to investigate their biological influence on biocompatibility and neuronal differentiation capacity, as well as the attenuation of inflammatory responses. Based on our results, the Pul–Col–Au nanocomposites exhibited both the best cell viability and anti-oxidative ability, as well as inhibition of the activation of the platelets and monocyte-macrophage conversion. Moreover, the motility of MSCs and HSFs was verified to be promoted through the highest expression of the CXCR4/SDF-1α axis and MMP activities. Furthermore, the results determined that the expression of nestin, β-tubulin, and GFAP was highly induced by Pul–Col–Au, indicating that the nanocomposites enhanced the neuronal differentiation for MSCs. Additionally, leukocyte infiltration, M1 polarization, capsule formation, and collagen deposition were all observed to be the lowest, as well as the TUNEL positive apoptotic cells, after subcutaneous implantation of Pul–Col–Au nanocomposites for one month. Finally, the high induction of M2 macrophage polarization and endothelialization also elucidated the potential for tissue regeneration. These findings have verified that Pul–Col–Au nanocomposites possess superior biocompatibility and the induction of neuronal differentiation capacity and can therefore be potential nanomaterials for use in tissue repair applications.

## Figures and Tables

**Figure 1 cells-10-03276-f001:**
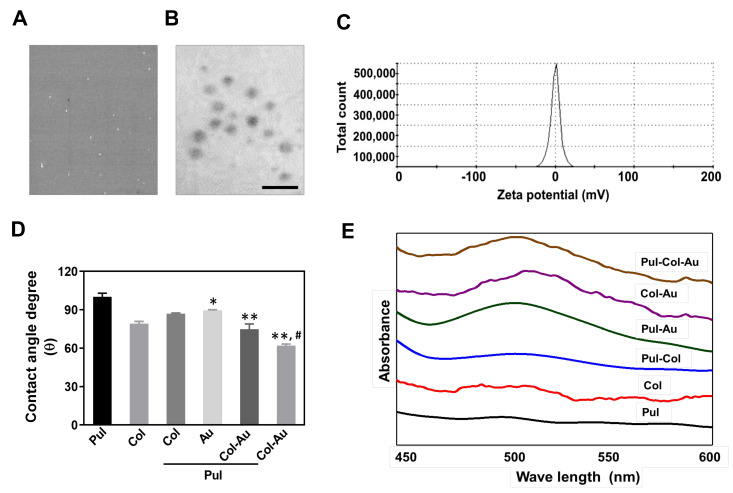
Characterization of pullulan-derived nanocomposites. (**A**) The SEM image of Pul–Col–Au nanoparticles. (**B**) The TEM image of Pul–Col–Au nanoparticles. The scale bar = 20 nm. (**C**) The zeta potential of Pul–Col–Au detected by DLS assay. (**D**) The water contact angle measurement of various nanomaterials, indicating that the degree (θ) of Pul–Col–Au and Col–Au nanocomposites was significantly lower than that of the control. The contact angle from different materials without water is θ = 0°. * *p* < 0.05, ** *p* < 0.01: compared to Pul group. ^#^
*p* < 0.05: compared to Col group. (**E**) UV-Vis spectra of various nanomaterials. The absorbance at 520 nm is the typical peak for the presence of Au nanoparticles. The wave-length of the spectra ranges from 450–600 nm. The data are represented as one of three independent experiments.

**Figure 2 cells-10-03276-f002:**
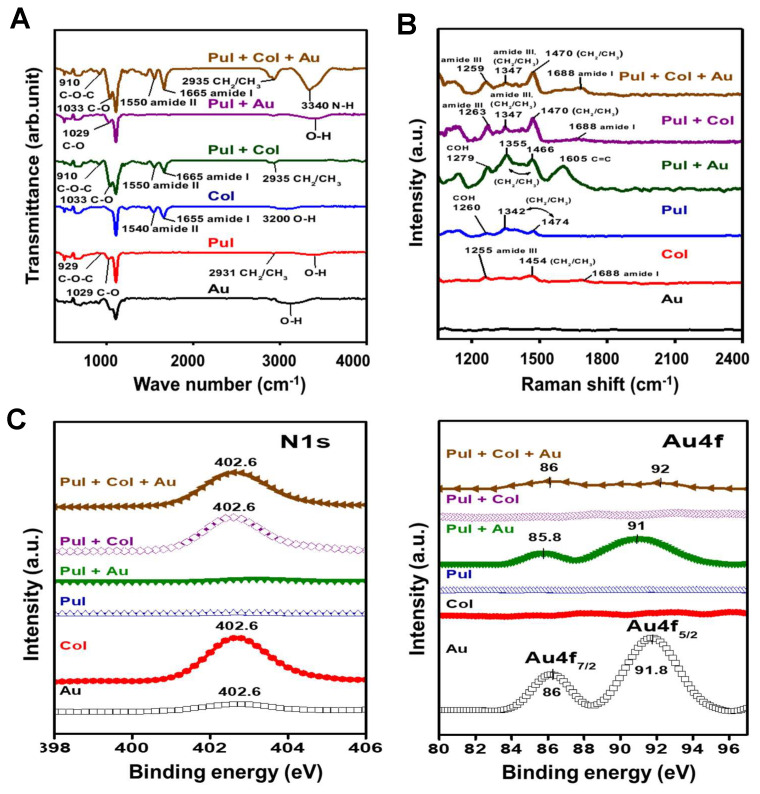
Identification of various nanomaterials by FTIR, SERS, and XPS assay. (**A**) The FTIR spectra evaluated the specific functional groups. The combination of Pul and Col led to the shift of C–O–C vibration from 929 cm^−1^ to 910 cm^−1^, C–O vibration from 1029 cm^−1^ to 1033 cm^−1^, CH_2_/CH_3_ functional group from 2931 cm^−1^ to 2935 cm^−1^, and amide II band from 1540 cm^−1^ to 1550 cm^−1^. N-H stretching at 3340 cm^−1^ was also found in the Pul–Col–Au group, indicating Au nanoparticles successfully bonding with Pul–Col. (**B**) The Raman shift spectra measured in the region from 1100 cm^−1^ to 2400 cm^−1^ by SERS. The specific absorption peaks of Pul–Col–Au were detected at 1259 cm^−1^ (amide III), 1347 cm^−1^ (CH_2_/CH_3_), 1470 cm^−1^ (CH_2_/CH_3_), and 1688 cm^−1^ (amide I), indicating the presence of Pul, Col, and Au. (**C**) The wide-scan XPS spectra of various nanomaterials. The combination of Pul and Col shows the N1s peak position at 402.6 eV binding energy due to the amide group in Col. The binding energy of Au at 86 and 91.8 eV was detected in Pul–Col–Au, indicating the structure of Pul–Col would not change by the addition of Au.

**Figure 3 cells-10-03276-f003:**
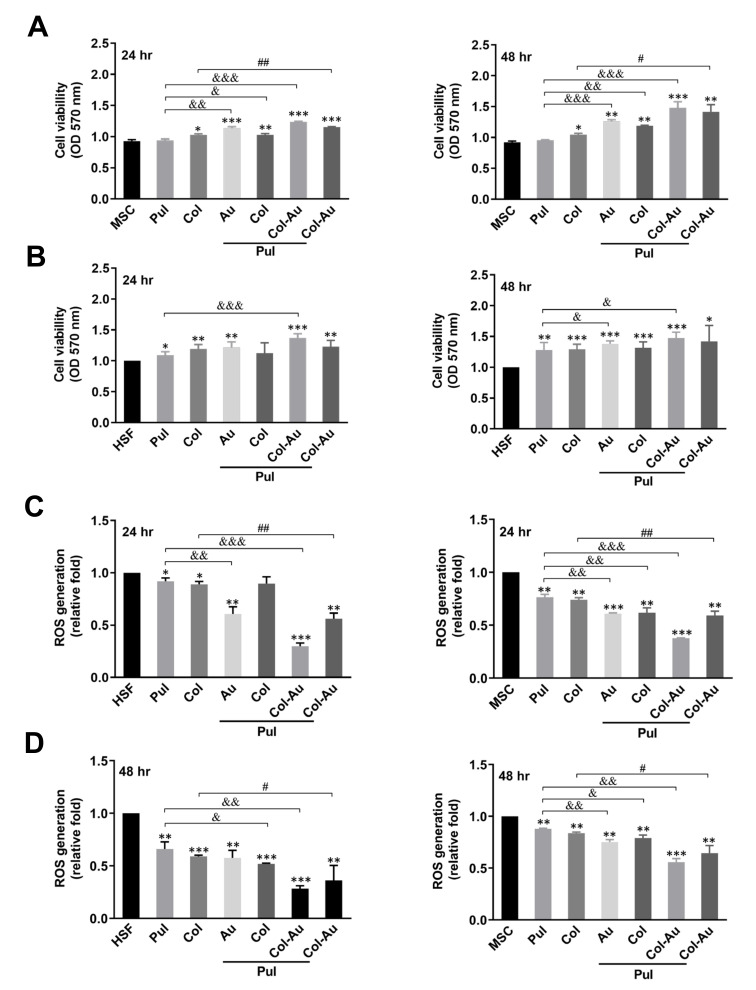
Biocompatibility assessments of MSCs and HSFs influenced by various materials. (**A**,**B**) MTT assay was applied to investigate the cell viability at 24 and 48 h. The results demonstrated the cell viability were the greatest for MSCs and HSFs in the Pul–Col–Au group at both time points. Data are represented as mean ± SD of the three independent experiments. (**C**,**D**) The intracellular ROS generation was detected by DCFH-DA and flow cytometry at 24 and 48 h. The semi-quantitative data indicates that Pul–Col–Au induced the lowest production in both HSFs and MSCs. Data are represented as mean ± SD of the three independent experiments. * *p* < 0.05, ** *p* < 0.01, *** *p* < 0.001: compared to MSC or HSF alone group. ^&^
*p* < 0.05, ^&&^
*p* < 0.01, ^&&&^
*p* < 0.001: compared to Pul group. ^#^
*p* < 0.05, ^##^
*p* < 0.01: compared to Col group.

**Figure 4 cells-10-03276-f004:**
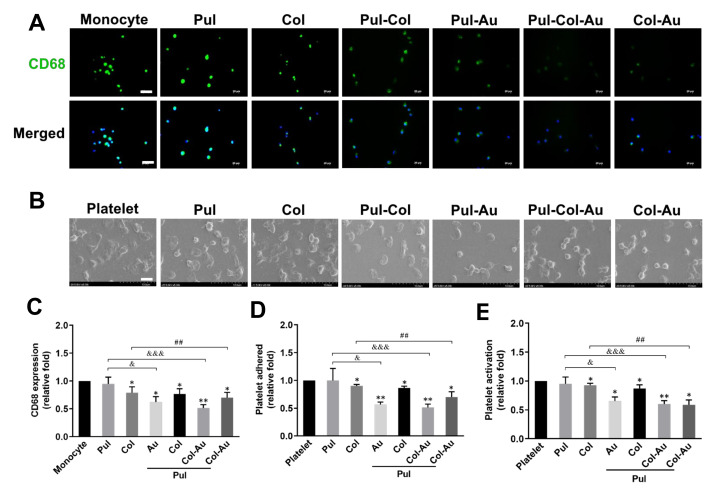
Effects of various materials in macrophage and platelet activation. (**A**) The macrophage marker CD68 (green color) was investigated by immunofluorescence staining at 96 h. DAPI was used to locate cell nuclei (blue color). Scale bar = 20 μm. (**B**) The morphology of platelets was observed by SEM analysis. Scale bar = 10 μm. (**C**) The CD68 positive cells were semi-quantified for macrophage activation. The results indicate the lowest expression was in Pul–Col–Au group. (**D**,**E**) The platelet adherence efficiency and degree of platelet activation were further quantified. The data also demonstrated that the lowest platelet adherence and activation was in the Pul–Col–Au group. Data are exhibited as mean ± SD of the three independent experiments. * *p* < 0.05, ** *p* < 0.01: compared to monocyte or platelet alone group. ^&^
*p* < 0.05, ^&&&^
*p* < 0.001: compared to Pul group. ^##^
*p* < 0.01: compared to Col group.

**Figure 5 cells-10-03276-f005:**
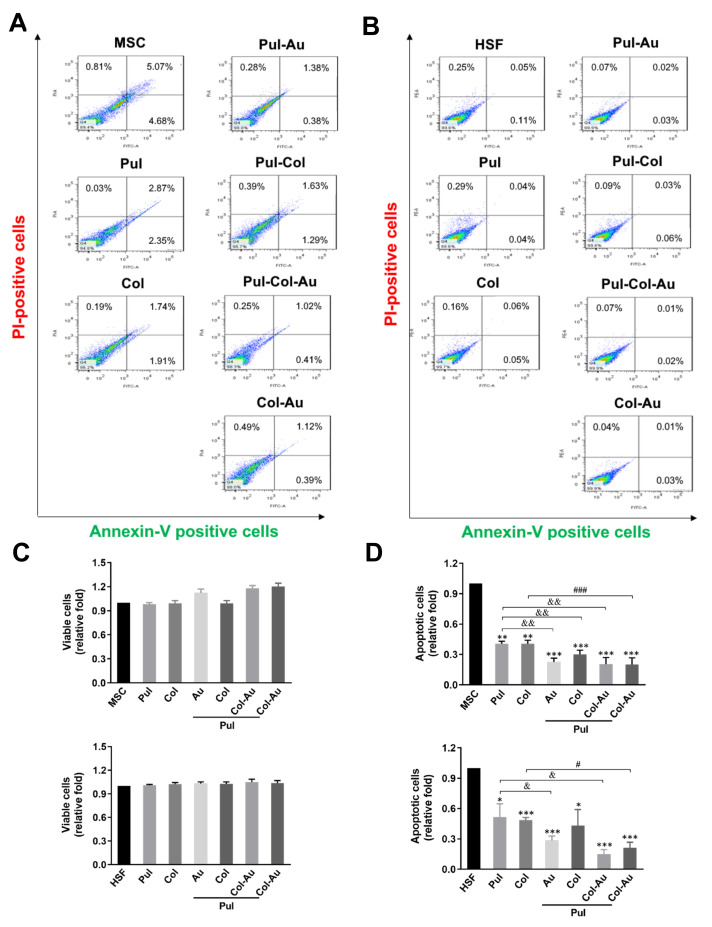
Apoptotic cells induced by various nanomaterials. Annexin V-PI double staining of (**A**) MSCs and (**B**) HSFs was investigated by flow cytometry at 24 h. (**C**) The quantitative results of the viable cells indicate there was no significant difference in the treated groups compared to the control for both cell types. (**D**) The quantitative data of the apoptotic cells demonstrated that the lowest amount was induced by the Pul–Col–Au group compared to the control for MSCs (~0.2-fold) and HSFs (~0.14-fold). Data are displayed as mean ± SD of the three independent experiments. * *p* < 0.05, ** *p* < 0.01, *** *p* < 0.001: compared to MSC or HSF alone group. ^&^
*p* < 0.05, ^&&^
*p* < 0.01: compared to Pul group. ^#^
*p* < 0.05, ^###^
*p* < 0.001: compared to Col group.

**Figure 6 cells-10-03276-f006:**
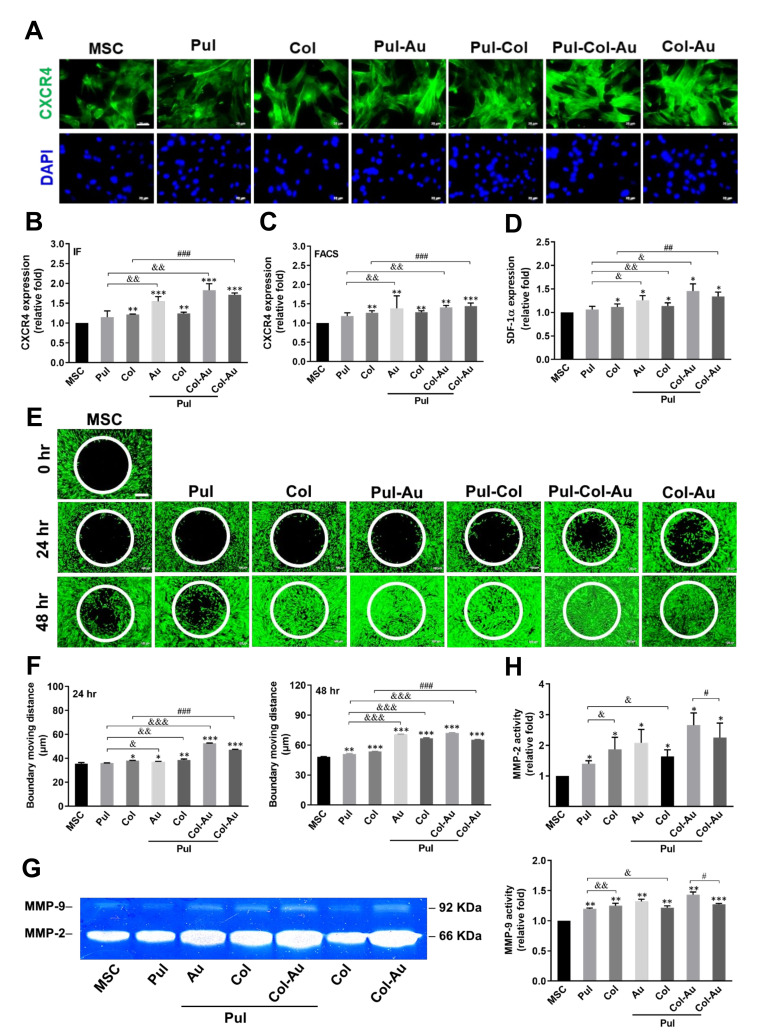
Assessment of biological function for MSCs culturing with various nanomaterials. (**A**) The immunofluorescence staining images of MSCs incubated with the anti-CXCR4 antibody (green color) and DAPI (blue color) at 48 h. Scale bar = 20 μm. (**B**) The semi-quantification results of CXCR4 expression based on fluorescence intensity. (**C**) The CXCR4 positive cells were also detected and analyzed by the FACS method. (**D**) The expression of SDF-1α protein was investigated and analyzed through ELISA assay at 48 h. (**E**) The real-time images of MSC migration influenced by different nanomaterials were stained with calcein-AM at 0 (as a reference), 24, and 48 h. Scale bar = 100 μm. (**F**) The boundary moving distance for each treated group was quantified, with the results indicating that the highest migration distance was in the Pul–Col–Au group (24 h: 52.5 μm, 48 h: 72 μm). * *p* < 0.05, ** *p* < 0.01, *** *p* < 0.001: compared to MSC alone group. ^&^
*p* < 0.05, ^&&^
*p* < 0.01, ^&&&^
*p* < 0.001: compared to Pul group. ^#^
*p* < 0.05, ^##^
*p* < 0.01, ^###^
*p* < 0.001: compared to Col group. (**G**) The zymogram of MMP activities for MSCs by gelatin zymography analysis at 48 h. (**H**) The semi-quantification of MMP-2/9 expression was evaluated by Image Pro Plus 5.0 software. The results also elucidate the greatest expression being in the Pul–Col–Au group (MMP-2: ~2.65-fold, MMP-9: ~1.43-fold). Data are expressed as mean ± SD of the three independent experiments. * *p* < 0.05, ** *p* < 0.01, *** *p* < 0.001: compared to MSC alone group. ^&^
*p* < 0.05, ^&&^
*p* < 0.01: compared to Pul group. ^#^
*p* < 0.05: compared to Pul–Col–Au group.

**Figure 7 cells-10-03276-f007:**
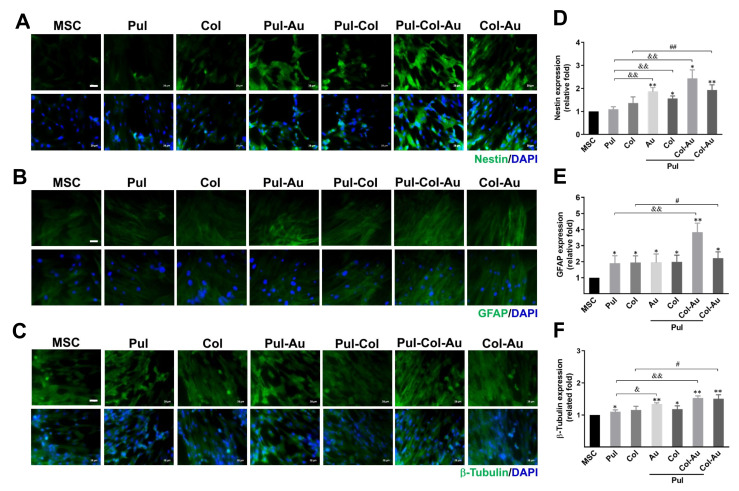
Identification of neuronal differentiation for MSCs influenced by various nanomaterials. The cells, after culturing with the nanomaterials for 7 days, were detected by immunostaining with neuronal markers (**A**,**D**) nestin, (**B**,**E**) GFAP, and (**C**,**F**) β-tubulin antibodies. The protein expression level was semi-quantified based on the fluorescent intensity, with the results indicating that the greatest expression for each marker was in the Pul–Col–Au group (nestin: ~2.43-fold, GFAP: ~3.83-fold, β-tubulin: ~1.52-fold). Scale bar = 20 μm. Data are represented as mean ± SD of the three independent experiments. * *p* < 0.05, ** *p* < 0.01: compared to MSC alone group. ^&^
*p* < 0.05, ^&&^
*p* < 0.01: compared to Pul group. ^#^
*p* < 0.05, ^##^
*p* < 0.01: compared to Col group.

**Figure 8 cells-10-03276-f008:**
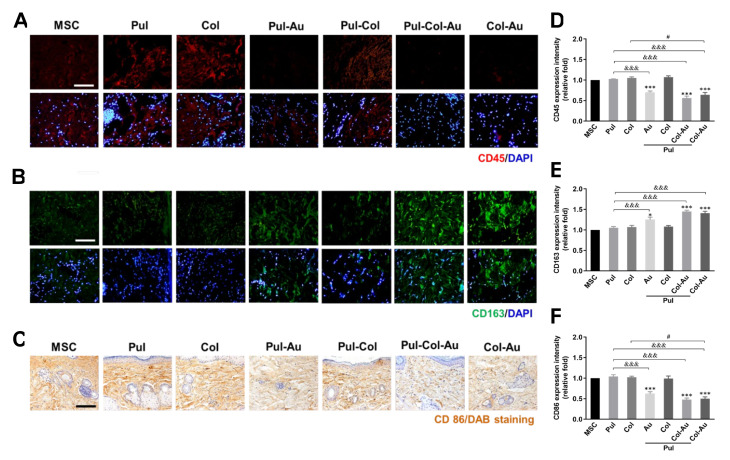
In vivo assessments for leukocyte filtration and macrophage polarization after implanting each material for one month. The immunohistochemistry images for (**A**) leukocyte filtration (CD45, red), (**B**) M2 polarization (CD163, green), and (**C**) M1 polarization (CD86, brown) by DAB staining. The expression of (**D**) CD45 and (**E**) CD163 was semi-quantified based on the fluorescent intensity. (**F**) The DAB positive cells for CD68 were analyzed and quantified the expression. The results indicate the lowest expression of CD45 (~0.56-fold) and CD86 (~0.48-fold), but the highest expression of CD163 (~1.45-fold). Scale bar = 100 μm. Cell nuclei were stained with DAPI (blue). Data are represented as mean ± SD of the three independent experiments. * *p* < 0.05, *** *p* < 0.001: compared to MSC alone group. ^&&&^*p* < 0.001: compared to Pul group. ^#^
*p* < 0.05: compared to Col group.

**Figure 9 cells-10-03276-f009:**
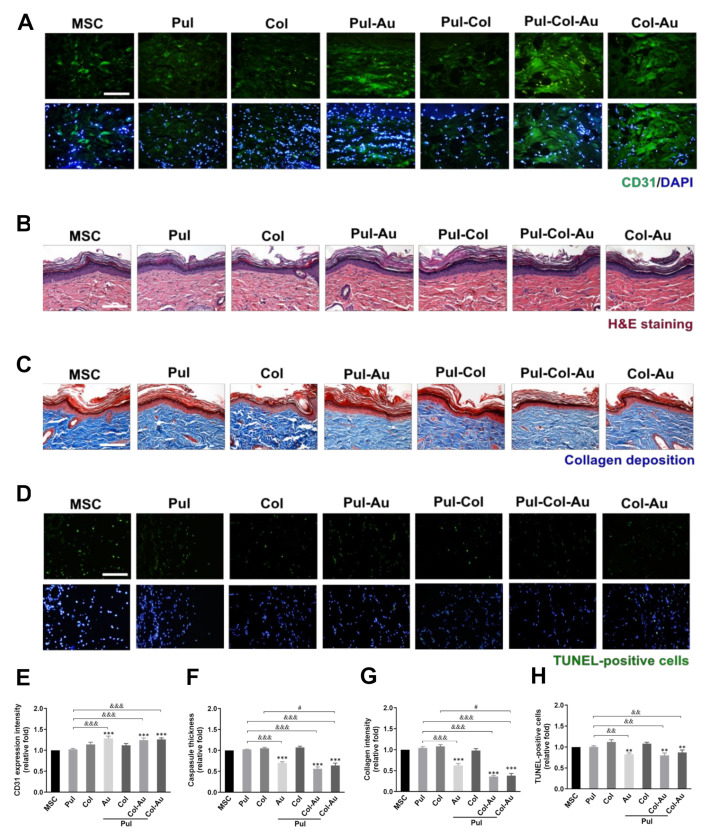
Foreign body responses induced by various materials after 4 weeks of implantation. (**A**) The images demonstrate the immunohistochemical analysis of endothelialization (CD31). Furthermore, the histological analysis of foreign body responses was processed. (**B**) Hematoxylin and eosin (H&E) staining for the capsule formation. (**C**) Masson’s trichrome-staining for the collagen deposition. (**D**) The apoptotic cells induced by various materials were also detected by TUNEL assay. (**E**) The quantified results of CD31 expression based on the fluorescent intensity, indicating the greater expression in the Pul–Col–Au (~1.24-fold) group. The semi-quantitative results of (**F**) capsule thickness, (**G**) collagen deposition intensity, and (**H**) TUNEL positive cells are demonstrated, indicating the lowest relative-fold for ~0.56, ~0.36, ~0.8 respectively. DAPI was used to locate cell nuclei (blue). Scale bar = 100 μm. Data are expressed as mean ± SD of the three independent experiments. ** *p* < 0.01, *** *p* < 0.001: compared to MSC alone group. ^&&^
*p* < 0.01, ^&&&^
*p* < 0.001: compared to Pul group. ^#^
*p* < 0.05: compared to Col group.

## Data Availability

Data are contained within the article.

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
