# Peer review of "Engineered Pullulan-Collagen-Gold Nano Composite Improves Mesenchymal Stem Cells Neural Differentiation and Inflammatory Regulation"

_cells, 2021, doi:10.3390/cells10123276_

Round 1
Reviewer 1 Report
- Please make the introduction more concise, particularly the part about previous treatments and their limitations, you can refer the readers to reviews about the wound therapies such as
https://journals.cambridgemedia.com.au/wpr/volume-26-number-2/advanced-wound-therapies.
It is better to focus more on what you have done i.e., stem cells and nanocomposites in the introduction.
- Please remove the part about TGF-β. It is out of the scope of your research.
- The sentence “Skin wounds can be caused by acute or chronic factors such as severe burn injury, diabetic ulcers or other mechanical injuries” is better be mentioned at the beginning of the introduction after skin properties and wounds description.
- What is the difference between Pul-Col-Au and Pul-Au-Col in the figure 3 graphs? I assume one of them does not have Pul but the line under them shows otherwise. Please modify the graph to avoid confusion.
- MSC characterization data needs to be included in the study. And could you justify the use of high passage number stem cells? High passage numbers can greatly influence MSC properties.
- Where is the origin of fibroblasts? If they are primary cells, the passage number is very high as well.
- Explain what & and # means on the graph bars.
- Again, what is the difference between the last two groups in the X-axis of Figure 4 C, D, E?
- Have you done any analysis on necrotic cells (Just PI) in figure 5?
- You should be consistent with the treatment group names. Somewhere you have called it Au-Col and somewhere else Col-Au.
- Figure 6 E; it looks like from images that the migration ability of cells on Col-Au is more than cells on Pul-Col-Au. However, graphs state otherwise. Which one is correct?
- What does (?) mean at line 704?
- The control group is mentioned with three different names on graphs (MSCs, Control, CTRL). Please be consistent.
- There are a couple of studies that have used Pullulan–Collagen or Pullulan- nanocomposites to Improve Wound Healing. You need to improve your literature review and include these studies in your discussion section. Some are below.
10.1089/ten.tea.2010.0298
https://doi.org/10.1016/B978-0-12-821280-6.00031-3
https://doi.org/10.1089/ten.tea.2020.0320
- Please use the abbreviated form of words after the first use.
- There is no discussion about the results on HSFs.
The manuscript has interesting findings; however, it needs a major revision. Figures need modifications to be more clear to readers. Improvement of discussion and introduction section is highly required. I hope my suggestions assist the authors to improve their manuscript.
Author Response
Comments and Suggestions for Authors
- Please make the introduction more concise, particularly the part about previous treatments and their limitations, you can refer the readers to reviews about the wound therapies such as https://journals.cambridgemedia.com.au/wpr/volume-26-number-2/advanced-wound-therapies. It is better to focus more on what you have done i.e., stem cells and nanocomposites in the introduction.
Answer:
Thanks for the valuable comment and suggestion from the Reviewer. We have reworded and included the new description in introduction section.
“Artificial materials combined with cells such as cell-based skin substitutes and epithelial/dermal replacements materials, are well applied in skin tissue engineering [16]. Thus, cell-based substitute comprising of living skin cells and ECM is considered to be more effective [17]. TransCyte is neodermis regeneration matrix invented by nylon mesh incorporating with allogeneic human dermal fibroblasts [18], and it is verified to facilitate re-epithelialization capacity and healing period for the patients suffering from partial or full-thickness skin burns [19]. Dermagraft is a degradable substitute produced by allogeneic human dermal fibroblasts and collagen scaffold, which shows tear resistance and decreases the infection for chronic Diabetic Foot Ulcers patients [20, 21]. Keratinocytes are the major cells in the process of re-epithelialization by secreting various cytolines and growth factors [22]. The treatments of keratinocytes dressing demonstrate no side-effects and facilitate wound repair for non-healing diabetic neuropathic foot ulcers [23]. However, the substitutes are commonly high cost, and need to storage under special condition [17].” (Page 2, line 81-93)
“Our previous studies demonstrated that MSCs culturing with nanogold-collagen or nanogold-fibronectin composites exhibited the enhancement of proliferation and endothelialization, indicating MSCs can be a fascinating material for tissue repair [26, 27].” (Page 2, line 98-101)
- Please remove the part about TGF-β. It is out of the scope of your research.
Answer:
Thanks for the suggestion from the Reviewer. We have removed the description about TGF-β in the introduction section.
- The sentence “Skin wounds can be caused by acute or chronic factors such as severe burn injury, diabetic ulcers or other mechanical injuries” is better be mentioned at the beginning of the introduction after skin properties and wounds description.
Answer:
Thanks for the valuable suggestion from the Reviewer. We have moved the sentence to the beginning of introduction section. (Page 2, line 49-52)
- What is the difference between Pul-Col-Au and Pul-Au-Col in the figure 3 graphs? I assume one of them does not have Pul but the line under them shows otherwise. Please modify the graph to avoid confusion.
Answer:
We have modified the mislabeling for the treatment group to the correct name “Pul-Col-Au” in each figure. The treatment groups in current study were “Pul, Col, Pul-Au, Pul-Col, Pul-Col-Au, Col-Au” respectively.
- MSC characterization data needs to be included in the study. And could you justify the use of high passage number stem cells? High passage numbers can greatly influence MSC properties.
Answer:
(1) Thanks for the valuable comment from the Reviewer. The results of MSC characterization were included in supplementary Figure S1.
“The specific surface markers of MSCs were firstly detected by a flow cytometry. The markers, CD34 and CD45, were highly expressed in endothelial cells and immune cells, which were represented as negative markers (0.87 % and 0.8 % respectively) (Figure S1A). The specific positive markers CD44 (99%) and CD90 (99.6%) for MSCs were remarkably detected (Figure S1B). Then the MSCs were applied in the following experiments.”
(2) We have confirmed the MSCs were at 8th passage in current study and included the description about MSC characterization in section 2.3.1. (Page 6, line 271-280) and section 3.2. (Page 11-12, line 526-530).
“To characterize the phenotypes of MSCs, the cells were harvested and detached with 2mM EDTA in phosphate-buffered saline (PBS), then washed by PBS with 2% bovine serum albumin (BSA) and 0.1% sodium azide (Sigma-Aldrich, Burlington, MA, USA). Further, the cells were incubated with antibodies conjugated with fluorescein isothiocyanate (FITC) or PerCP-Cy5-5-A. The indicated markers were CD34-FITC, CD45-FITC, CD44-PE, and CD90-PerCP-Cy5-5-A (BD Pharmingen, San Diego, CA, USA). FITC-conjugated IgG1 and PerCP-Cy5-5-A-conjugated IgG1 (BD Pharmingen) were demonstrated as isotype controls. The specific surface antigens of MSCs were detected by a flow cytometry [42]. Then, the cells were analyzed by FACS software (Becton Dickinson LSR II, Canton, MA, USA). The 8th passage of MSCs were used in the current study.” (Page 6, line 271-280)
- Where is the origin of fibroblasts? If they are primary cells, the passage number is very high as well.
Answer:
(1) Human fibroblasts used in current study were purchased from American Type Culture Collection (ATCC).
(2) Our previous study described the HSFs can be cultured to higher passages, and do not lose cell growth rate and phenotypes. We also included the description in section 2.3.2.
“Human skin fibroblasts were obtained from American Type Culture Collection (ATCC). HSFs can be cultured to higher passages without losing proliferation rate and cell phenotypes, which was described in our previous study [43].” (Page 6, line 283-285)
- Explain what & and # means on the graph bars.
Answer:
Thanks for the suggestion from the Reviewer. We have included the explanation for the meanings of “& and #” in each figure legend.
- Again, what is the difference between the last two groups in the X-axis of Figure 4 C, D, E?
Answer:
We have modified the name of treatment groups to be consistent in each figure.
- Have you done any analysis on necrotic cells (Just PI) in figure 5?
Answer:
We thank the valuable comment from the reviewer. Based on widely study, necrotic cells are specific for permeable for PI, which intercalates in nuclear DNA and is visible as red fluorescence. Applications propidium Iodide makes it possible to identify and quantitate dead and necrotic cells on a single-cell basis by flow cytometry. Therefore, we just staining necrotic cells by using PI.
- You should be consistent with the treatment group names. Somewhere you have called it Au-Col and somewhere else Col-Au.
Answer:
Thanks for the suggestion from the Reviewer. We have checked the name of each treatment group and modified them to be consistent in figures and figure legends.
- Figure 6 E; it looks like from images that the migration ability of cells on Col-Au is more than cells on Pul-Col-Au. However, graphs state otherwise. Which one is correct?
Answer:
We have changed the real-time images to the correct states and make sure the images can correspond to Figure 6F. (Page 17)
- What does (?) mean at line 704?
Answer:
We have removed the “(?)” and included “the deposition” in the sentence. (Page 19, line 767)
- The control group is mentioned with three different names on graphs (MSCs, Control, CTRL). Please be consistent.
Answer:
Thanks for the suggestion from the Reviewer. We have renamed the control group to be “MSC” in each figure to make them be consistent.
- There are a couple of studies that have used Pullulan–Collagen or Pullulan- nanocomposites to Improve Wound Healing. You need to improve your literature review and include these studies in your discussion section. Some are below.
Answer:
Thanks for the valuable suggestion from the Reviewer. We have included several literatures related to pullulan nanocomposites in section Discussion. (Page 22, line 874-888
“Several studies have described the Pul-based nanocomposites for promoting wound healing. The aminoalkysilane was grafted on bacterial nanocellulose (BNC) membrane to form A-g-BNC, then combined with fabricated pullulan-zinc oxide (Pul-ZnO) to generate A-g-BNC/Pul-ZnO nanoparticles. A-g-BNC/Pul-ZnO exhibited better biocompatibility for L929 fibroblast cells, and the animal assessments elucidated the enhancement of re-epithelialization, formation of blood vessels which is contributed to wound repair [69]. A piece of literature has verified the curcumin grafted hyaluronic acid modified pullulan polymer (Cur-HA-SPul) could facilitate L929 cells proliferation and anti-microbial capacity. The rat model results also figured out the promotion of wound healing [70]. The novel three dimensional film, chitosan/carboxymethyl pullulan polyelectrolyte complex (PEC) loaded with 45S5 bioglass (CCMPBG), was reported to enhance mechanical strength and biodegradation behavior for wound tissue regeneration [71]. Additionally, a research team developed the hydrogel comprising of Pul-Col nanocomposites and adipose-derived stem cells (ASC), and the hydrogel induced better re-epithelialization, vascularization, and expression of angiogenesis-related genes VEGF and SDF-1α [72].”
- Please use the abbreviated form of the words after the first use.
Answer:
Thanks for the valuable suggestion from the Reviewer. We have reworded the abbreviated form of the words.
- There is no discussion about the results on HSFs.
Answer:
Thanks for the suggestion from the Reviewer. We have included the discussion about HSFs in discussion section. (Page 21, line 827-836)
“HSFs have been well applied for skin repair applications. Previous literature described a cellularized bilayer skin substitute comprising of pullulan-gelatin hydrogel (PG-1), primary HSFs, and keratinocytes. PG-1 provided biocompatible environment for the the bilayer created by HSFs and keratinocytes, and further promoted the formation of thicker neo-dermis [33]. Furthermore, researchers had created Pul-Col composite hydrogels to investigate in vitro biocompatibility. The human foreskin fibroblasts exhibited significant viability (> 97%) after culturing with Pul-Col hydrogels as well as demonstrated better invasion and attachment [37]. The above evidence figured out the combination of Pul-based nanocomposites and HSFs providing confidence for wound healing (Figure 3B & S3B).”

Reviewer 2 Report
This paper describes that the nanoparticles composed of pullulan, collagen, and gold nanoparticles have biocompatibility, neuronal differentiation capacity, and attenuation of inflammatory responses. Therefore, they have a potential for use in tissue repair applications. Actually, the authors investigated many aspects and effects of the nanoparticles by using various assays in vitro and even in vivo. Overall, I feel this manuscript contains a number of interesting new observations and merits publication, but the authors should consider the following issues for further improvements.
- The abstract is too long. Please, shorten it to less than 200 words (according to the Cells guideline).
- Results 3.1: What is the size of Pu-Co-Au? Is the size larger than gold particles (3-5 nm)?
- The composite ratio of Pu, Co, and Au in the particle should be described.
- In vivo experiments, the nanoparticles were coated on the bottom surface of 96-well plates just incubating with them. How many of the nanoparticles were coated per area?
- Line 300: For the SEM preparation, after dehydration of alcohol, the fixed cells may be deformed. In addition, is it possible to observe the samples by SEM without metal coating?
- Line 397: Please, explain how to treat the nanoparticles in the implant, and how much of the nanoparticles was added?
- Line 419: ix71 —> IX71
- Figure 3A: The symbols $, &&&, and #, which appear in many other graphs, may be mistaken? In addition, in the x-axis, Col-Au may be Au-Col.
- Results 3.2: Please, add the doubling time in the growth of MSC and HSF. Did the cell number increase only about 1.5 folds after 48 h-culture?
- Figure 4B: Scale bar is missing.
- Results 3.3: The authors should describe how much of the population in the control showed apoptosis.
- Figure 6B-D: Did the authors examine the averaged fluorescence intensities of stained cells or count the number of stained cells? How much of the population was stained with the antibody? Please, make it clear.
- Figure 6E: In this type of experiment, not only cell migration but cell growth also affects the closure speed of the black region. Tracking individual cells may be better.
- I could not find the supplementary figures Fig S1-S4.
- Results 3.6: The roles of CD86, CD163, and CD45 are not clear here. I found this explanation in Discussion (Line 794). Please, move this explanation prior to the experiments in the Results.
- Because several pieces of research similar to this paper have been published including the authors’ papers, I want to see some discussions to compare with previous reagents.
Author Response
This paper describes that the nanoparticles composed of pullulan, collagen, and gold nanoparticles have biocompatibility, neuronal differentiation capacity, and attenuation of inflammatory responses. Therefore, they have a potential for use in tissue repair applications. Actually, the authors investigated many aspects and effects of the nanoparticles by using various assays in vitro and even in vivo. Overall, I feel this manuscript contains a number of interesting new observations and merits publication, but the authors should consider the following issues for further improvements.
- The abstract is too long. Please, shorten it to less than 200 words (according to the Cells guideline).
Answer:
Thanks for the suggestion from the Reviewer. We have shortened the abstract section. (Page 1, line 24-39)
“Tissue repair engineering supported by nanoparticles and stem cells has been demonstrated as being an efficient strategy for promoting the healing potential during the regeneration of damaged tissues. In the current study, we prepared various nanomaterials including pure Pul, pure Col, Pul-Col, Pul-Au, Pul-Col-Au and Col-Au to investigate physicochemical properties, biocompatibility, biological functions, differentiation capacities and anti-inflammatory abilities through in vitro and in vivo assessments. The physicochemical properties were characterized by SEM, DLS assay, contact angle measurements, UV-Vis spectra, FTIR spectra, SERS and XPS analysis. The biocompatibility results demonstrated Pul-Col-Au enhanced cell viability, promoted anti-oxidative ability for MSCs and HSFs, as well as inhibited monocyte and platelet activation. Pul-Col-Au also induced the lowest cell apoptosis and facilitated the MMP activities. Moreover, we evaluated the efficacy of Pul-Col-Au in the enhancement of neuronal differentiation capacities for MSCs. Our animal models elucidated better biocompatibility, as well as the promotion of endothelialization after implanting Pul-Col-Au for a period of one month. The above evidence indicates the excellent biocompatibility, enhancement of neuronal differentiation and anti-inflammatory capacities, suggesting that the combination of pullulan, collagen and Au nanoparticles can be potential nanocomposites for neuronal repair, as well as skin tissue regeneration in any further clinical treatments.”
- Results 3.1: What is the size of Pu-Co-Au? Is the size larger than gold particles (3-5 nm)?
Answer:
In this study, a FESEM was used to observe the Pu-Co-Au composite. Use Image pro software to analyze the size of the particle on the image. The result is that the diameter of the particle is 3.8 ± 0.2 mm (n=10) and larger than gold nanoparticle. The above results show that the pullulan and collagen molecular chains are entangled around the gold nanoparticle.
The new description was included in section 2.2.1. (Page 5, line 213-214) and section 3.1. (Page 9, line 463-465).
- The composite ratio of Pu, Co, and Au in the particle should be described.
Answer:
Thanks for your comment. The description “The weight percentages of pullulan, collagen and gold in Pul-Col-Au composite are 99.9986%, 0.00020% and 0.00119%, respectively.” is included into section 2.1.5. (Page 4, line 201-202)
- In vivo experiments, the nanoparticles were coated on the bottom surface of 96-well plates just incubating with them. How many of the nanoparticles were coated per area?
Answer:
Thanks for your comment.
(1) It is known that the particle size of gold is 5 nm. It can be calculated that the volume of each gold nanoparticle (AuNP) is 1.9635Í10-13/cm2. The density of gold is 19320 mg/cm3. It can be calculated that the weight of a AuNP is 3.79348Í10-9 mg. When preparing the composite material, 244 ul of 12.2 ppm AuNP solution was added, and finally the whole was diluted to 10 ml. It can be calculated that the solution contains 0.0029768 mg of AuNP, which can be converted into 78,471 AuNPs /ml. Take 0.1ml of the above solution and add it to a 96-well plate for plating. Therefore, it can be calculated that each coating contains 7847 AuNP. The area of the 96-well plate is 0.32 cm2, so each square centimeter contains 24522 AuNPs. The coating density is 245223 nanoparticles per square centimeter, and each 96-well plate has 78471 nanoparticles.
(2) In vivo experiments, Pu-Co-Au composite solution was coated on glass coverslip (15 mm) and implanted to rat subcutaneous tissue. It is known that the particle size of gold is 5 nm. It can be calculated that the volume of each gold nanoparticle (AuNP) is 1.9635Í10-13/cm2. The density of gold is 19320 mg/cm3. It can be calculated that the weight of a AuNP is 3.79348Í10-9mg.
“When preparing the composite material, 244 ul of 12.2ppm AuNP solution was added, and finally the whole was diluted to 10 ml. It can be calculated that the solution contains 0.0029768 mg of AuNP, which can be converted into 78,471 AuNP/ml. Take 0.594 ml of the above solution and add it to a glass coverslip for plating. Therefore, it can be calculated that each coating contains 149096 AuNPs. The area of the glass coverslip is 1.9 cm2, so each square centimeter contains 78,471 AuNP.” (Page 4, line 195-200)
- Line 300: For the SEM preparation, after dehydration of alcohol, the fixed cells may be deformed. In addition, is it possible to observe the samples by SEM without metal coating?
Answer:
Thanks for your comments. In order to avoid cell morphology deformed. General research will use glutaraldehyde to fix the cell morphology. To avoid the dehydration procedure leading to changes in cell morphology, dehydrated in the ethanol solutions of slow increasing concentrations (30%, 50%, 60%, 70%, 80%, 90%, and 100ï¼…) are used. On the other hand, the under sentence, sputter-coated with gold, was added in Line 327.
- Line 397: Please, explain how to treat the nanoparticles in the implant, and how much of the nanoparticles was added?
Answer:
Thanks for your comment. “In this study, gold nanoparticle solution was coated on glass coverslip (15 mm) and implanted to rat subcutaneous tissue. It can be calculated that the volume of each gold nanoparticle (AuNP) (~ 5 nm) is 1.9635Í10-13 /cm2. The density of gold is 19320 mg/cm3. It can be calculated that the weight of a AuNP is 3.79348Í10-9mg. When coated on glass coverslip (15 mm), add 552 ml of 50 ppm AuNP solution on the glass coverslip. It can be calculated that the solution contains 0.0028mg of AuNP, which can be converted into 72,787 AuNP on the coating.” (Page 8-9, line 424-430)
- Line 419: ix71 —> IX71
Answer:
Thank you for the correction from the Reviewer. We have revised “ix71” to “IX71”. (Page 9, line 452)
- Figure 3A: The symbols $, &&&, and #, which appear in many other graphs, may be mistaken? In addition, in the x-axis, Col-Au may be Au-Col.
Answer:
Thanks for the valuable suggestion from the Reviewer.
- We have included more information about the symbol “& and #” in each figure legend.
- We have checked and renamed the treatment groups to be consistent. The name of each treatment group is “Pul, Col, Pul-Au, Pul-Col, Pul-Col-Au, Col-Au” respectively.
- Results 3.2: Please, add the doubling time in the growth of MSC and HSF. Did the cell number increase only about 1.5 folds after 48 h-culture?
Answer:
(1) The doubling time of HSF and MSC different materials after 48 hr of incubation was 21.72 hr and 23.54 hr. Because MSCs belong to stem cells, the characteristic of proliferation ability was general slower than on mature cells such as HSFs. The number of cells was increased about 1.5 times after 48 hr of incubation.
doubling time=txlog2/(logN-logN0)
|
HSF |
MSC |
||
|
Day 0 |
Day 2 (48 hr) |
Day 0 |
Day 2 (48 hr) |
|
10000 |
15230 |
10000 |
14500 |
|
Doubling time (HSF) culture: 21.72 hr |
Doubling time (MSC) culture: 23.54 hr |
||
- Figure 4B: Scale bar is missing.
Answer:
Thanks for the valuable comment from the Reviewer. We have added the scale bar in Figure 4B.
- Results 3.3: The authors should describe how much of the population in the control showed apoptosis.
Answer:
Thanks for the valuable suggestion from the Reviewer. We have included the apoptosis population in control group for both MSCs and HSFs. “Moreover, the results seen in Figure 5D indicate that the apoptotic cell population of MSCs was lowest in both the Pul-Col-Au (~ 0.2 fold) and Col-Au (~ 0.2 fold) groups, followed by the Pul-Au group (~ 0.22 fold), Pul-Col group (~ 0.3 fold), pure Col group (~ 0.4 fold), pure Pul group (~ 0.4 fold), and MSC alone group (1 fold). The apoptotic population of HSFs was also the lowest in Pul-Col-Au group (~ 0.14 fold), followed by Col-Au group (~ 0.21 fold), Pul-Au group (~ 0.28 fold), Pul-Col group (~ 0.43 fold), pure Col group (~ 0.48 fold), pure Pul group (~ 0.51 fold), and HSF alone group (1 fold).” (Page 15, line 639-646)
- Figure 6B-D: Did the authors examine the averaged fluorescence intensities of stained cells or count the number of stained cells? How much of the population was stained with the antibody? Please, make it clear.
Answer:
We appreciate the valuable comment from the reviewer. We have co-stained the F-actin in MSCs. However, the merged fluorescence images were not displayed in order to avoid the interference. The images were showed as below. The fluorescence expression of CXCR4 demonstrated the co-localization with DAPI. Thus, the results elucidated that MSCs were verified to be CXCR4-positive cells.
- Figure 6E: In this type of experiment, not only cell migration but cell growth also affects the closure speed of the black region. Tracking individual cells may be better.
Answer:
We appreciate the valuable comment from the reviewer. The dynamic migration of MSC on different materials need recorded by a real-time cultured-cell monitoring system (CCM-Multi, Astec, Japan). Unfortunately, experimental equipment used for tracking individual cells was not available in our country now due to COVID-19 pandemic. However, our previous evidences confirmed this assay was available for detecting the phenomenon of migration effect due to did not cause any destroy the integrity of various nano films in the culture plate.
References:
(1) Huey-Shan Hung, Cheng-Ming Tang, Chien-Hsun Lin, Shinn-Zong Lin, Mei-Yun Chu, Wei-Shen Sun, Wei-Chien Kao, Hsieh-Hsien Hsu, Chih-Yang Huang, Shan-hui Hsu*, Biocompatibility and favorable response of mesenchymal stem cells on fibronectin-gold nanocomposites, PLoS One, 2013, 8(6): e65738.
(2) Huey-Shan Hung, Chih-Hsuan Chang, Chen-Jung Chang, Cheng-Ming Tang, Wei-Chien Kao, Shinn-Zong Lin, Hsieh Hsien-Hsu, Mei-Yun Chu, Wei-Shen Sun, Shan-hui Hsu*, In vitro study of a novel nanogold-collagen composite to enhance the mesenchymal stem cell behavior for vascular regeneration, PLoS One, 2014, 9(8): e104019.
- I could not find the supplementary figures Fig S1-S4.
Answer:
We have included the new supplementary figures in the revision version.
- Results 3.6: The roles of CD86, CD163, and CD45 are not clear here. I found this explanation in Discussion (Line 794). Please, move this explanation prior to the experiments in the Results.
Answer:
Thanks for the suggestion from the Reviewer. We have included the information for CD86, CD163, and CD145 in section 3.6. and make it to be more clear explanation.
“Leukocytes infiltration would be induced by the M1 polarization macrophages [45].” (Page 18, line 745-746)
“M2 polarization macrophages contributed to inhibit inflammation and strengthen angiogenesis [46], which were evaluated by CD163 IHC staining” (Page 19, line 750-752)
“Furthermore, the CD86 marker of M1 polarization macrophages induced the pro-inflammatory cytokines expression [46]” (Page 19, line 754-755)
- Because several pieces of research similar to this paper have been published including the authors’ papers, I want to see some discussions to compare with previous reagents.
Answer:
Thanks for the valuable suggestion from the Reviewer. We have included several references to compared the reagents in Discussion section. (Page 22-23, line 889-902)
“Our recent studies also demonstrated various nanocomposites for tissue regeneration. The component of extracellular matrix, fibronectin (FN), was fabricated with silver nanoparticles (AgNPs). AgNPs was verified to own the anti-bacterial ability. In vitro and in vivo measurements figured out FN-AgNP significantly inhibited the expression of pro-inflammatory cytokines (TNF-α, IL-1β and IL-6), promoted endothelialization for MSCs, and attenuated foreign body responses, demonstrating the potential vascular repair applications of FN-AgNPs [73]. Graphene-oxide decorated with Au nanoparticles (Go-Au) were suggested to be promising nanocarrier owing to better biocompatibility and inducing various differentiated cell types for MSCs such as neuron, endothelial cells, osteocytes and adipocytes [74]. The synthetic polymer, polyethylene glycol (PEG), was combined with Au nanoparticles. The results indicated the surface modification of Au nanoparticles on the PEG film significantly improved biological performance and biocompatibility [75]. The above evidences indicated nanoparticles such as gold and silver could enhance the functions of biomaterials.”

Reviewer 3 Report
In this paper, the authors reported the fabrication of Pul-Col-Au nanocomposites and investigated its functionality with both in vitro and in vivo study. This is a well-designed study with decent amount of data supplied. From the perspective of academic criticism, several technical concerns need to be addressed to further improve the quality of this manuscript, as appended below.
The SEM image is not so informative. TEM would be a better technique for the characterization.
For the biocompatibility test, the cell viability in long term culture (≥ 96hr) should be included.
Figure 6B. The semi-quantitative measurement should be based on the positive cell number/DAPI cell number ratio but not fluorescent intensity.
Please supply the flow gating for Figure 6C.
Semi-quantitative measure on band intensity should be included for Figure 6G.
Other than the IF staining of neural differentiation markers, a IF staining of stemness marker (e.x., SOX2) should be included to demonstrate the change on MSC properties.
Author Response
In this paper, the authors reported the fabrication of Pul-Col-Au nanocomposites and investigated its functionality with both in vitro and in vivo study. This is a well-designed study with decent amount of data supplied. From the perspective of academic criticism, several technical concerns need to be addressed to further improve the quality of this manuscript, as appended below.
- The SEM image is not so informative. TEM would be a better technique for the characterization.
Answer:
Thanks for the valuable suggestion from the Reviewer. We have included the TEM image and figure legend for the nanoparticles, which are represented as Figure 1B (Page 10).
And the new description of experimental method was also included in section 2.2.2.
“2.2.2. Transmission Electron Microscope (TEM)
TEM images of the nanoparticles were obtained from transmission electron microscope (JEM 1010, JEOL Ltd., Akishima, Tokyo, Japan). The voltage was set at 80 keV to clearly observe the size and structure of the nanoparticles. Before the observation, 5 μl of nanoparticle was suspended onto copper-coated TEM grid and dried out at room temperature.” (Page 5, line 216-221)
- For the biocompatibility test, the cell viability in long term culture (≥ 96hr) should be included.
Answer:
Thanks for the suggestion from the Reviewer. We have included the 96 hr culture of cell viability for both MSC and HSF in supplementary Figure S3. The new description was also included in section 3.2. (Page 12, line 568-576)
“Additionally, the cell viability in long term culture (96 hour) for both MSCs and HSFs was examined. The results for MSC at 96 hours was significantly higher in Col-Au, Pul-Col-Au and Pul-Au groups (OD570nm=1.84, 1.8, 1.76, respectively), followed by Pul-Col, MSC alone, pure Pul, and pure Col groups (OD570nm=1.64, 1.62, 1.6, 1.58, respectively) (Figure S3A). The data for HSF at 96 hours was remarkably greater in Pul-Au, Pul-Col-Au groups, and Col-Au (OD570nm=1.86, 1.85, 1.84, respectively), followed by pure Col, HSF alone, Pul-Col, pure Pul groups (OD570nm=1.8, 1.79, 1.78, 1.76, respectively) (Figure S3B). The evidence indicated the cell viability of Pul-Col-Au group was significantly higher in both MSC and HSF.”
- Figure 6B. The semi-quantitative measurement should be based on the positive cell number/DAPI cell number ratio but not fluorescent intensity.
Answer:
We appreciate the valuable comment from the reviewer.
We have co-stained the F-actin in MSCs. However, the merged fluorescence images were not displayed in order to avoid the interference. The images were showed as below. The fluorescence expression of CXCR4 demonstrated the co-localization with DAPI. Thus, the results elucidated that MSCs were verified to be CXCR4-positive cells.
Scale bar=20μm.
- Please supply the flow gating for Figure 6C.
Answer:
Thanks for the valuable suggestion from the Reviewer. We have included the flow gating results for each treatment group in the Supplementary Figure 5 in order to support the data of Figure 6C.
“The flow gating results were also demonstrated in Figure S5 to support the above FACS data.” (Page 16, line 668-669)
- Semi-quantitative measure on band intensity should be included for Figure 6G.
Answer:
The semi-quantitative results of band intensity were displayed as Figure 6H.
- Other than the IF staining of neural differentiation markers, a IF staining of stemness marker (e.x., SOX2) should be included to demonstrate the change on MSC properties.
Answer:
Thanks for the comment from the Reviewer. The data of MSC markers was included in Supplementary Figure1. We identified the specific surface marker of MSCs (CD44, CD90) by flow cytometry instead of the common marker Sox-2 for stem cells. The positive markers of CD44 and CD90 were over 99% expression, thus, the MSCs were confirmed to maintain stem cell phenotypes.

Round 2
Reviewer 2 Report
The authors properly responded to my comments.
I found 1 mistake.
Line 465: 0.2 µm should be 0.2 nm.
Author Response
Thanks for the comment. We have corrected the mistake in line 465.

Reviewer 3 Report
The reviewer thank the authors for commuting science in a good and efficient. The manuscript should be accepted for publication.
Author Response
We have thanks and appreciate for the valuable comment from reviewer.